# Huntingtin (HTT) interactome in regulation of DNA repair/remodeling and RNA processing pathways

Tamara Ratovitski[1] , Chloe D Holland[1], Robert N O'Meally[2], Alexey V Shevelkin[1], Siddhi V Kamath[1], Tianze Shi[1] ,
Matthew J Rodriguez[1] , Robert N Cole[2], Mali Jiang[1], Christopher A Ross[1,3]

**Huntington's disease (HD), an uncurable neurodegenerative disorder, is caused by CAG repeat expansion in the *HD* gene encoding mutant huntingtin protein. DNA damage response is implicated in HD pathogenesis. We used multiple approaches to assess normal and mutant HTT interactomes in the context of genotoxic stress. We show that double-strand break (DSB) repair response is impaired in HD neurons, which are more vulnerable to DSB-induced stress. We found that S1181 phosphorylation of HTT is regulated by DSB, and can be carried out by DNA-PK. Functional interaction of HTT with a major DSB kinase DNA-PKcs and association of both proteins with nuclear speckles suggest a role of HTT in DSB repair mechanism; however, physiological outcome of these interactions remains to be examined. We revealed HTT interactions with other proteins associated with nuclear speckles, TCERG1 and MED15, whose loci are genetic modifiers for HD, and with chromatin remodeling complex BAF. These interactions may position HTT as an important scaffolding intermediary providing integrated regulation of gene expression and RNA processing in the context of DNA repair mechanisms.**

## Introduction

Huntington's disease (HD) is a progressive neurodegenerative disorder with no disease-modifying therapies currently available (1, 2, 3, 4). It is caused by a single mutation, CAG repeat expansion in the HD gene (*HTT*) and message (HTT), coding for expanded polyglutamine (polyQ) near the N terminus of the mutant huntingtin (mHTT) protein (5). There is preferential degeneration of medium spiny neurons in the striatum, especially early in the disease course. HTT is a ~350 kD protein with suggested scaffolding function (6) and numerous protein interactions, many of which are altered in HD (7, 8, 9, 10, 11, 12, 13, 14). The mechanism of HD

pathogenesis is not completely understood, and it is likely to involve abnormal interactions of the mHTT protein.

Many studies have implicated the HTT protein in the cellular DNA damage response (DDR) (15, 16), but most attention was concentrated on mismatch repair pathways because several of the proteins involved in DNA mismatch repair (FAN1, MSH2) are encoded by genes located at HD modifier loci (17, 18, 19, 20, 21, 22, 23). However, the gene products of mismatch repair–related genes have not emerged as HTT interactors yet and may be involved in HD pathogenesis through other mechanisms. Other studies suggest that HTT is involved in double-strand break (DSB) repair mechanisms. The mutant HTT interacts with Ku70, an essential DSB repair protein and a regulatory component of the DNA-dependent protein kinase (DNA-PK) (24, 25). HTT was found at DNA damage sites colocalizing with another major DSB-induced kinase, ATM (ataxia telangiectasia mutated) (26), and targeting ATM-ameliorated mutant huntingtin toxicity (27). HTT has also been implicated in transcription-coupled (TC) DNA DSB repair coordinating a chromatin remodeler BRG1(SMARCA4)-dependent TC nonhomologous end-joining (TC-NHEJ) complex to repair DSB in neurons (16, 28, 29 *Preprint*).

RNA processing has also emerged as a molecular mechanism associated with HD, based on several expression profiling, RNA-seq, and proteomics studies in HD models (11, 13, 30, 31, 32, 33, 34, 35). Our previous methyl-proteome analysis has identified a cluster of RNA-binding splicing factors interacting with HTT (35). Many RNA-binding proteins (RBPs), including those interacting with HTT, have been recently implicated in the DDR (36, 37, 38, 39, 40, 41, 42). HTT role in chromatin remodeling via its interaction and facilitation of polycomb repressive complex (PRC2) have been suggested (43). However, although there are many reports of HTT-interacting proteins, to our knowledge there has not been a systematic investigation of the HTT interactome under conditions of DNA damage stress.

An extensive somatic expansion of the *mHTT* CAG tract (up to ~100 CAG) was shown to take place in medium spiny neuron (MSN) populations that are selectively vulnerable in HD, and it is

[1]Division of Neurobiology, Department of Psychiatry and Behavioral Sciences, Johns Hopkins University School of Medicine, Baltimore, MD, USA    [2]Mass Spectrometry and Proteomics Facility, Department of Biological Chemistry, Johns Hopkins University School of Medicine, Baltimore, MD, USA    [3]Departments of Neurology, Neuroscience and Pharmacology, Johns Hopkins University School of Medicine, Baltimore, MD, USA

Correspondence: tratovi1@jhmi.edu; caross@jhu.edu
Alexey V Shevelkin's present address is Confocal and Electron Microscopy Core, National Institute on Drug Abuse, National Institutes of Health, Baltimore, MD, USA

accompanied by HD-associated gene expression changes (44). The most recent studies from the McCarroll laboratory combining CAG repeat sizing with RNA-seq in the same individual cells have suggested that MSNs have far more expansions to very long repeat lengths (150+ CAG) than other neuron types, and that this is central to HD pathogenesis (45). They showed that MSNs with up to 150 CAG repeats have surprisingly few changes in gene expression, whereas neurons with greater than 150 CAGs have loss of important MSN-appropriate, cell type–defining messages and inappropriate expression of messages related to development or cell toxicity. Our immortalized striatal precursor neuron (ISPN) HD model (with 180 CAG and expanding) (46) is uniquely suited to study altered interactions of the mHTT protein with very long repeats as most relevant to HD pathogenesis.

Although recent studies suggest that somatic expansion may be an important driver of pathology (44, 45), this is likely to be just one component of the HD pathogenic mechanism. The most recent GWAS identified several HD modifiers not obviously involving DNA mismatch repair (MED15, RRM2B, CCDC82, and TCERG1) (47). The same study found that noncanonical *HTT* CAG repeat sequences (CAA.CCA) modify motor onset but do not increase *HTT* CAG repeat expansion, suggesting a different mechanism from the initial somatic expansion phase.

Using the ISPN model, we now find that DSB repair response is impaired in HD neurons and that HTT phosphorylation at the S1181 site is regulated by the induction of DSBs. We show that phospho-HTT interacts with DNA-PKcs (catalytic subunit of DNA-PK complex) and that both proteins colocalize to nuclear speckles (NS), essential subnuclear organelles involved in all steps of RNA processing. In addition, we found HTT interaction with components of chromatin remodeling complex BAF.

Our analysis of the HTT interactome using affinity purification and a pilot study, testing a novel proximity-based approach, supports the function of HTT in the regulation of translation, RNA metabolism, chromatin remodeling, and DSB repair (distinct from mismatch repair), suggesting potential role of HTT in facilitating interplay and emerging interconnections between these pathways.

# Results

### DSB repair response is impaired in HD immortalized striatal precursor neurons (ISPNs)

One of the earliest consequences of activation of DDR kinases (ATM, DNA-PK, and ATR) at DSBs is phosphorylation of the histone variant H2A.X on serine (S139) producing γH2AX, which is a key step in signaling and initiating the repair of DSBs. After the initial phosphorylation of H2A.X at sites flanking DSBs, numerous DSB repair proteins are recruited resulting in further activation of ATM and phosphorylation of H2A.X histones, forming γH2A.X foci that are easily detected by immunofluorescence-based assays (48). Thus, γH2A.X is a sensitive molecular marker of DSB repair (49). We sought to evaluate DSB repair response in HD neurons by measuring and quantifying γ-H2A.X foci that are formed in our immortalized striatal precursor neurons (ISPNs) (46) undergoing

DDR. ISPNs are human patient-derived induced pluripotent stem cells (iPSCs) differentiated to a striatal precursor stage and then immortalized. DSB/DDR was induced by treatment with bleomycin (5–10 µg/ml, 30 min), which led to the activation of γ-H2A.X and p-ATM as shown by immunoblotting (Fig 1A). γ-H2A.X foci were visualized by immunofluorescence (IF) with γ-H2A.X–specific antibody (Fig 1B). As described before (50), small γ-H2A.X foci are formed at the early stages of the DDR, decreasing in number and increasing in size as the DDR progressed. It has been suggested that when the number of DSBs exceeds the cellular capacity, γH2A.X is visualized as "apoptotic ring" or diffuse halos rather than distinct foci, and these are associated with apoptosis and not with DSB (50). We observed such pan-nuclear γH2A.X staining in about 5–10% of the cells (Fig 1C), and these were excluded from analysis.

We employed a 3D visualization and quantitation using Imaris software. Nuclei were chosen as a region of interest for individual cells, and 3D surface was constructed using the "DAPI" channel for quantitation of nuclear γ-H2A.X foci. Our analysis shows significantly lower γH2A.X focus size and intensity in HD ISPNs compared with the control (Fig 1D), consistent with a deficiency of HD ISPNs in mounting an adequate DSB repair response. HD ISPNs demonstrate more vulnerability to DSB-induced (by bleomycin) stress compared with normal neurons, as measured by decreased viability (ATP) and increased apoptosis (activation of caspase 3/7, Fig 1E).

### Phosphorylated huntingtin participates in DSB/DDR via interaction with components of nuclear speckles (NS)

Posttranslational modifications (PTMs) are major regulators of protein function and interactions. Several PTMs of HTT have been found to modulate HTT biochemical properties and expanded HTT toxicity (6, 51, 52, 53, 54, 55, 56, 57). We sought to investigate whether the stoichiometry of HTT phosphorylation at specific sites is affected by bleomycin-induced DNA damage, thus suggesting a role of these modifications in the regulation of DSB/DDR pathways in HD. Using parallel reaction monitoring (PRM), we quantified by mass spectrometry the relative stoichiometry of phosphorylation sites in HTT from control and HD ISPNs with and without bleomycin treatment. We targeted the most prominent phosphorylation sites previously identified in various systems (57). Unmodified peptides from HTT were used for normalization to account for HTT in samples. Phosphorylation at S421, S1201, S1876, and S2114/2116 sites was significantly increased in 180CAG ISPNs compared with control cells in both stressed and nonstressed conditions (Fig S1). The stoichiometry of phosphorylation at one site, S1181, was dramatically increased in HD cells upon bleomycin treatment (Fig 2A) suggesting potential role of this PTM in DSB response in HD ISPNs.

To further investigate the details of subcellular localization of pS1181-HTT, we employed IF with previously validated (57, 58) (see the Materials and Methods section for details) phospho-specific antibody to the S1181 site followed by confocal microscopy. We observed localization of endogenous pS1181 HTT within the nuclear compartment, forming nuclear puncta (Fig 2B). 3D visualization and quantitation analysis (Imaris) showed significantly increased volume and intensity of these puncta in HD ISPNs after induction of

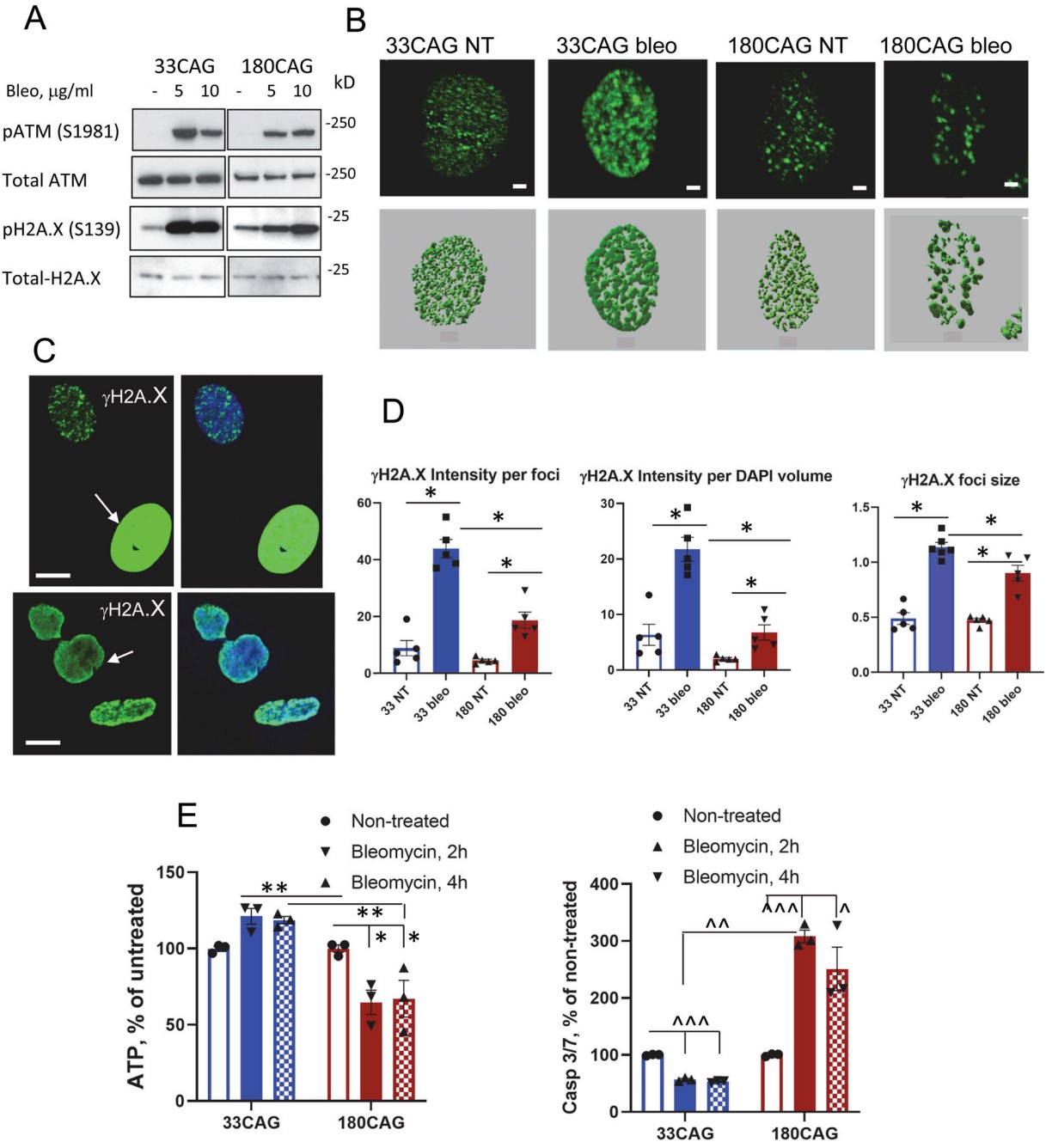

**Figure 1. Double-strand break DNA damage response (DSB/DDR) is impaired in HD.**
**(A)** DSB/DDR induction by bleomycin is measured by γ-H2A.X and p-ATM immunoblotting. **(B)** Representative images (top panels) and 3D reconstruction (bottom panels, Imaris) of control and HD ISPNs stained with γ-H2A.X–specific antibody upon induction of DSB by bleomycin (10 μg/ml, 30 min) or from untreated cells (NT). Scale bar, 2 μm. **(C)** Pan-nuclear γH2A.X staining of apoptotic cells, not associated with DSB. Examples of apoptotic cells (~5–10%) not included in analysis. IF staining with γ-H2A.X–specific antibody. Nuclei were visualized with DAPI. Scale bar, 10 μm. **(D)** 3D quantification (Imaris) of γH2A.X foci. Data represent the mean ± SEM. Y-axis labels are displayed at the top of each graph. Two-sample t tests with equal variances were performed. *P < 0.05, n = 5 (five images taken from independent wells with 7–12 cells per image). DSB/DDR quantification shows lower γH2A.X focus size and intensity in HD ISPNs compared with normal ISPNs. **(E)** HD ISPNs are more vulnerable to DSB-induced stress. Control (33CAG) and HD (180CAG) ISPNs were treated with 10 μg/ml of bleomycin for the indicated time, and ATP and Casp3/7 levels were measured. Data are presented as the mean % ± SEM of the corresponding nontreated group. One-way ANOVA with pairwise multiple comparison procedures (Holm–Sidak's method): *P = 0.029, **P = 0.001, n = 3. t test with equal variances: ^P < 0.017, ^ ^P = 0.004, ^ ^ ^P = 0.001, n = 3.

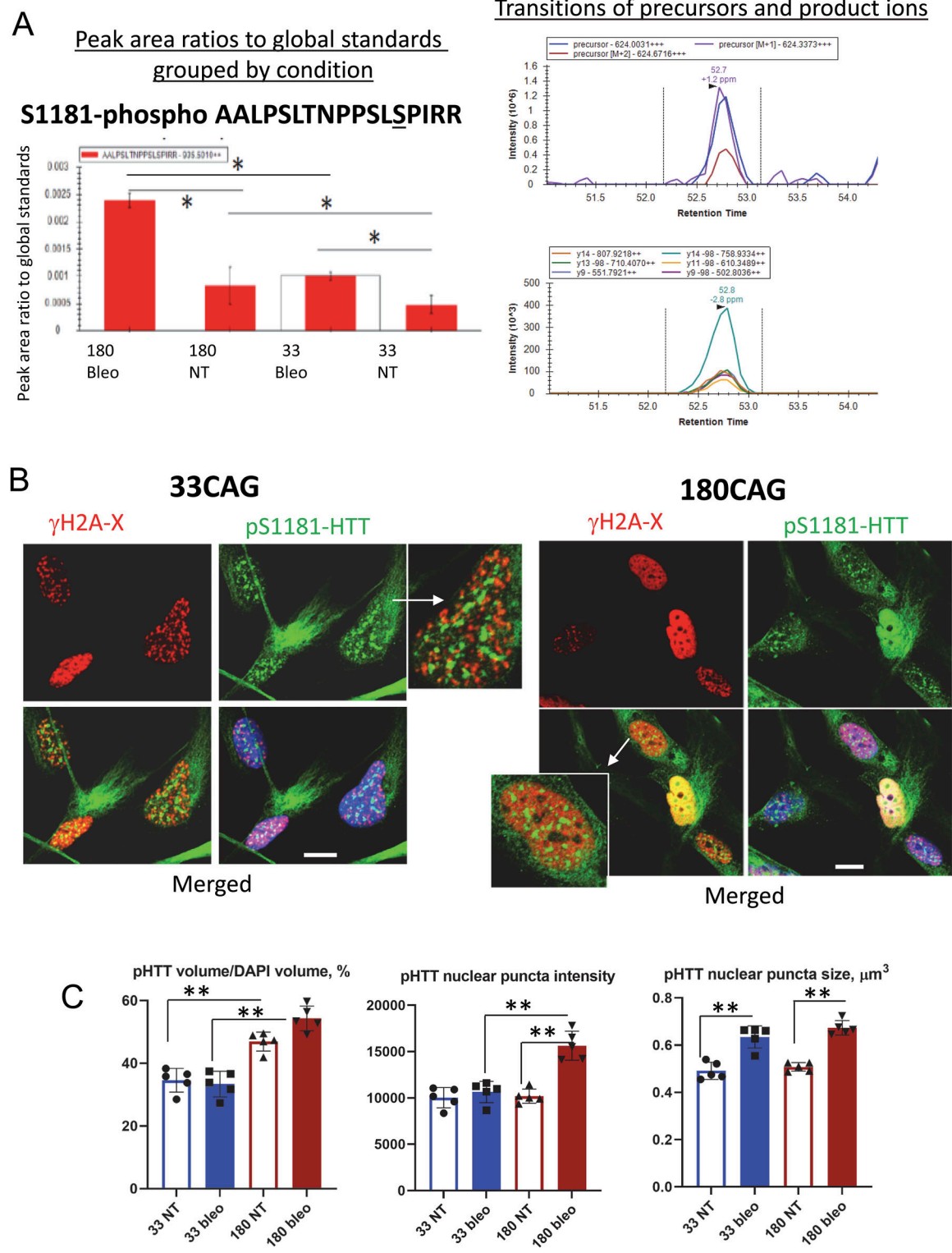

**Figure 2. HTT phosphorylation at S1181 is modulated by the induction of DSB response.**

**(A)** Relative quantification of phospho-serine S1181 by parallel reaction monitoring (PRM) mass spectrometry. PRM measured abundance of AALPSLTNPPSLpSPIRR peptide from normal or polyQ-expanded mHTT proteins from HTT bands cut out from the gel after HTT immunoprecipitation from total cell lysates of control (33CAG) and HD (180CAG) ISPNs upon induction of DSB by bleomycin (10 μg/ml, 30 min), or from untreated cells (NT). PRM data were analyzed with the Skyline program. Graphs show peak area ratios to global standards (unmodified peptides from HTT) grouped by the condition. Statistical data analysis was performed by the Skyline program using triplicate samples for each cell line (n = 3). *P < 0.05. P-value was adjusted for the multiple comparisons within this group. The adjustment was done using the Benjamini–Hochberg procedure by the Skyline program. Transitions (m/z ratio of a peptide and its corresponding product ion m/z) are shown on the right. **(B, C)** pS1181 HTT is localized to nuclear puncta with increased size and intensity (in HD ISPNs) after induction of DSB. **(B)** Representative images of normal and HD ISPNs treated

DSB, compared with normal ISPNs (Fig 2C). Notably, pS1181 HTT puncta did not colocalize with γH2AX foci in control or HD ISPNs (Fig 2B) and were also negative for other subnuclear markers: coilin (Cajal bodies) and PML (PML nuclear bodies) (Fig S2). Instead, we found a striking colocalization with nuclear speckle (NS) marker SC35/SRSF2 (Fig 3A). NS are subnuclear organelles involved in all aspects of RNA processing (59). Mean Manders' colocalization coefficient reflecting co-occurrence of two channels (representing HTT and SC35) was 0.9, whereas mean Pearson's coefficient, which measures the correlation of intensities between two channels, was greater than 0.7, suggesting a high degree of colocalization. There was no significant difference between normal (33CAG) and HD (180CAG) neurons (Fig 3B). As assessed by 3D quantitation (Fig 3C), organization of SC35 into nuclear speckles was altered in HD cells. Intensity, density, and total volume of SC35-positive nuclear speckles increased upon induction of DSB in normal ISPNs. In contrast, these measures were decreased or not changed in HD cells. NS positive for both phospho-HTT and SC35 appeared disorganized and completely disintegrated in a proportion of HD ISPNs with a few large speckles formed in some cells (indicated by arrows in Fig 3A).

We sought to test whether HTT may interact with other proteins associated with NS in our ISPN system. We targeted TCERG1 (transcription elongation regulator 1), a coilin-associated factor essential for Cajal body formation and integrity, and snRNP biogenesis (60). The *TCERG1* locus is also a genetic modifier for HD (47, 60, 61), and the TCERG1 protein has previously been reported to interact with HTT in yeast two-hybrid screens and in vitro assays (62, 63) and to colocalize with the NS marker SC35 (64). Using confocal microscopy with orthogonal projections of z-stacks, we observed TCERG1/HTT partial colocalization in the nuclei of ISPNs (Fig 4A). Mean Manders' colocalization coefficient reflecting co-occurrence of HTT and TCERG1 channels was greater than 0.9, whereas mean Pearson's coefficient (which measures the correlation of intensities between two channels) was greater than 0.7 in normal ISPNs suggesting a high degree of colocalization. Both coefficients were significantly reduced in HD (180CAG) neurons suggesting reduced colocalization (Fig 4B). To confirm endogenous interaction of HTT with TCERG1, we conducted co-immunoprecipitations (co-IPs) in our ISPN model (Fig 4C). Total cell lysates were prepared from control (33) and HD (180) ISPNs treated with bleomycin (10 μg/ml, 2 h) and untreated (NT), and HTT complexes were immunoprecipitated using antibodies to total HTT (MCA2050). TCERG1 and HTT were detected in the immunoprecipitates (IPs). Proximity ligation assay (PLA) showed robust PLA signals in the nuclei of both untreated and bleomycin-treated normal (33CAG) ISPNs, suggesting endogenous interactions (Fig 4D and E). A dramatic decrease in PLA site number and intensity was observed in HD ISPNs, with some induction upon bleomycin treatment, suggesting decreased interaction in HD cells. (PLA site measurements were reported relative to technical negative single

antibody controls. Representative images of negative controls are shown in Fig S3A).

## Unbiased screen for HTT interactors upon induction of DSB/DDR

Diminished DSB response observed in HD neurons, and HTT interaction with a major RNA processing organelle (NS) prompted us to explore how HTT may be involved in the regulation of interconnected DNA damage response and RNA processing pathways. We conducted an unbiased HTT interactome screen in ISPNs differentiated into medium spiny neuron–like cultures (as described previously), undergoing genotoxic stress. Differentiated ISPNs (46) recapitulate major HD-related phenotypes of the parental iPSC model, including brain-derived neurotrophic factor (BDNF) withdrawal–induced cell death that can be rescued by small molecules previously validated in the parental iPSCs (7, 8, 13). After induction of DSBs with bleomycin (10 μg/ml, 30 min), HTT protein complexes were pulled down from normal (33CAG) and HD (180CAG) neurons using a mixture of anti-HTT antibodies (MCA2050 and 2051) followed by quantitative mass spectrometry (TMT-based) to compare the relative protein abundance in the pulldowns. Three samples were analyzed for each condition. 469 proteins have been identified with at least two unique peptides with high confidence (1% FDR) in all six HTT pulldowns (Table S1). Protein abundance was normalized to HTT protein abundance to account for variability between individual HTT pulldowns. Out of these proteins, 331 decreased >20% in abundance in HD ISPNs (HD/Control < 0.8), whereas 40 increased by >20% (HD/control > 1.2). Biological replicates of two groups of cells were well separated by the principal component analysis (PCA) based on normalized protein abundance (generated using PD2.4, Fig 5A). The hierarchical clustering heat map (Fig 5B) and volcano plots (Fig 5C) demonstrate that most of the proteins were less abundant in the pulldowns from HD neurons.

We focused on proteins found less abundant in the pulldowns from HD ISPNs (relative to control cells) and performed pathway and process enrichment analysis using "Metascape" analysis tools (65). This dataset was significantly and highly enriched in biological processes associated with protein translation and RNA metabolism (mRNA splicing and micro-RNA biogenesis, Fig 5D). Protein–protein interaction (PPI) enrichment analysis (Metascape) of the same dataset identified protein networks with best-scoring terms by *P*-value with functional description of the corresponding networks shown in the table in Fig 5E. Networks involved in translation, chromatin remodeling, and RNA processing have been identified with the highest enrichment *P*-value among proteins less abundant in the pulldowns from HD ISPNs suggesting potential dysregulation of these pathways because of weakened or partially lost interactions with HTT in HD neurons. These results are consistent with previous expression profiling, RNA-seq, and proteomics studies in HD models implicating RNA biogenesis,

---

with bleomycin (10 μg/ml, 30 min) costained with pS1181-HTT and γ-H2A.X–specific antibodies. Nuclei were visualized with DAPI. Scale bar, 10 μm. **(C)** 3D quantification (Imaris) of pS1181 HTT nuclear puncta. Data represent the mean ± SEM. Y-axis labels are displayed at the top of each graph. One-way ANOVA with pairwise multiple comparison procedures (Holm–Sidak's method): **$P < 0.001$, n = 5 (five images taken from independent wells with 7–12 cells per image).

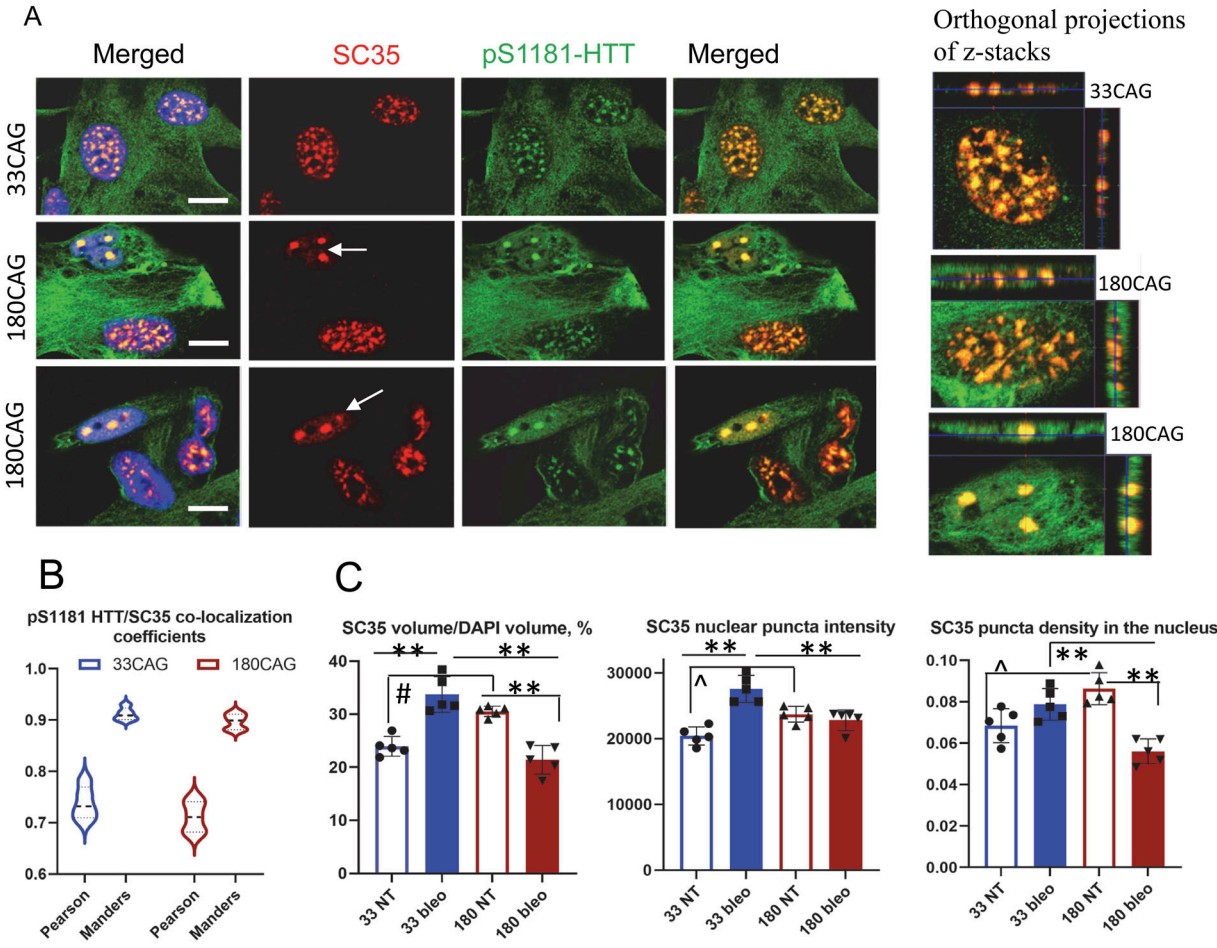

**Figure 3. Phospho-HTT colocalizes with nuclear speckle marker SC35.**
**(A)** Representative images of ISPNs treated with bleomycin (10 µg/ml, 30 min) costained with pS1181-HTT and SC35 antibodies. Nuclei were visualized with DAPI. Orthogonal projections of Z-stacks (right panels) demonstrate colocalization. Scale bar, 10 µm. **(B)** Graph shows Pearson's and Manders' colocalization coefficients. The data are presented as the mean ± SD. No significant difference between normal (33CAG) and HD (180CAG) ISPNs was observed (*t* test with equal variances). n = 5 (five images taken from independent wells with 5–10 cells per image; cells were analyzed individually, and means of coefficients were calculated for each image). **(C)** 3D quantification (Imaris) of SC35-positive nuclear speckles (NS) in normal (33) and HD (180) ISPNs. Data represent the mean ± SEM. Y-axis labels are displayed at the top of each graph. One-way ANOVA with pairwise multiple comparison procedures (Holm–Sidak's method): \*\**P* < 0.001, #*P* = 0.002, ^ *P* = 0.006, n = 5 (five images taken from independent wells with 7–12 cells per image).

processing, splicing, and gene expression in HD pathogenesis (11, 13, 30, 31, 32, 33).

### HTT is phosphorylated by and interacts with DNA-PKcs

We targeted DNA-PKcs (PRKDC), one of the proteins that showed (by quantitative mass spectrometry) decreased abundance in HTT pulldowns from HD ISPNs (HD/control = 0.64, Fig 6A). DNA-PKcs is a catalytic subunit of an essential DSB repair complex, DNA-dependent protein kinase (DNA-PK). DNA-PK is central in the process of nonhomologous end joining (NHEJ), which is the main DSB repair pathway used by neurons (reviewed in references 16, 48, 66). The mutant HTT has been previously shown to interact with Ku70 (24, 25), a regulatory component of this complex required for recruitment and activation of DNA-PKcs (67).

We asked whether DNA-PK can phosphorylate HTT. In vitro phosphorylation assays were conducted with the recombinant HTT-23Q/HAP40 complex (purified by Curia as described in the Materials and Methods section). The recombinant DNA-PK complex (purified from HeLa cells, Promega), activated by the addition of linear double-stranded DNA, was incubated with HTT/HAP40 for indicated times, and the phosphorylation state of HTT was assessed using previously described and validated (57, 58) phospho-specific antibodies to four phosphorylation sites on HTT (see the Materials and Methods section for details). We observed a significant increase in S1181 phosphorylation (normalized to total HTT), but not at other sites analyzed (Fig 6B).

We confirmed endogenous interaction of HTT with DNA-PKcs in our ISPN model using co-IP (Fig 6C). Total cell lysates were prepared from control (33CAG) and HD (180CAG) ISPNs treated with bleomycin (10 µg/ml, 2 h) and untreated (NT), and HTT complexes were immunoprecipitated using antibodies to total HTT (MCA2050, two top panels). DNA-PKcs and HTT were detected in the IPs. Notably, an antibody to DNA-PKcs was able to pull down pS1181-

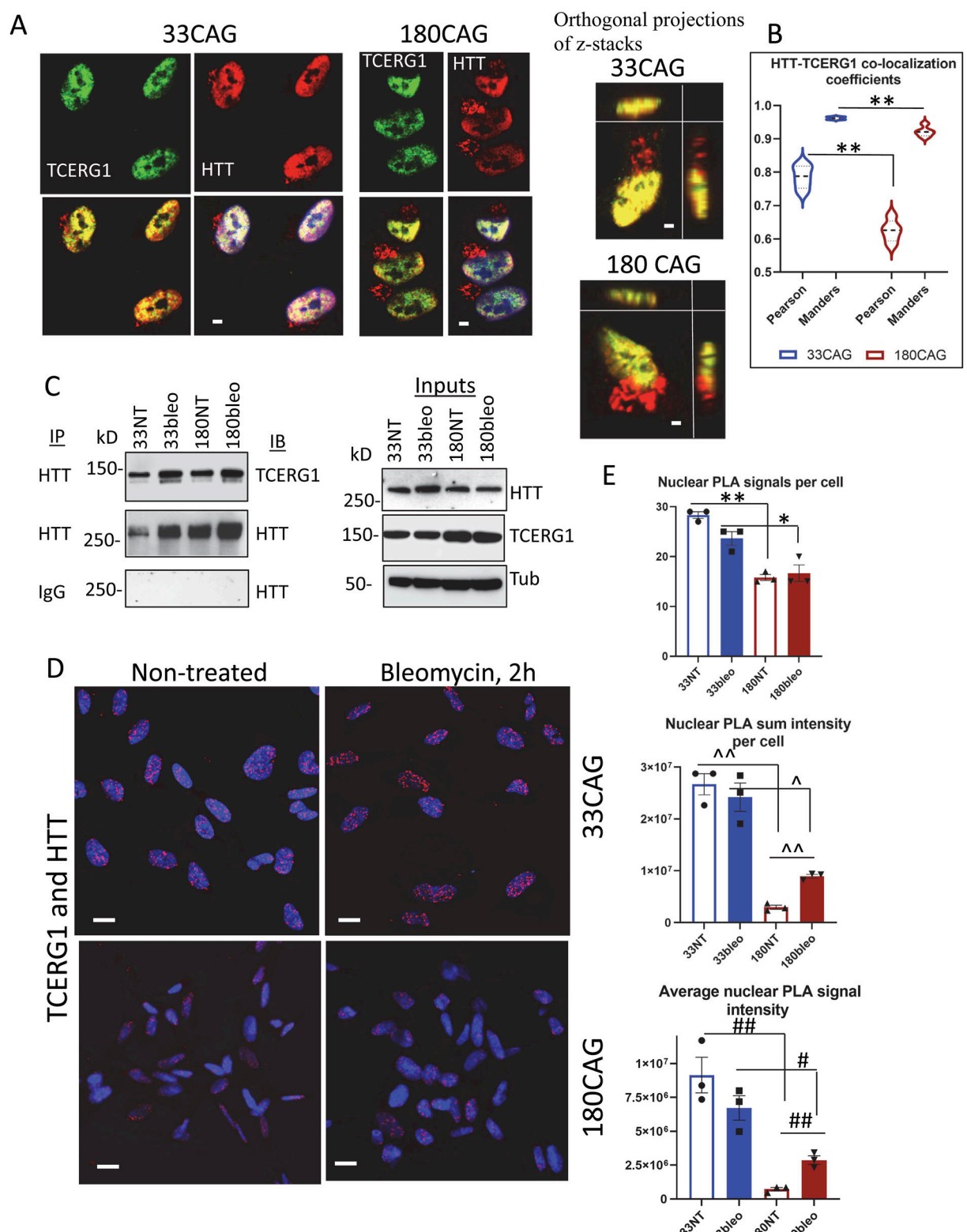

**Figure 4. HTT/TCERG1 colocalization and interaction in ISPNs.**
**(A)** Representative images of ISPNs treated with bleomycin (10 μg/ml, 2 h) costained with TCERG1 and HTT (MCA2050) antibodies. Orthogonal projections of z-stacks (right panels) demonstrate colocalization. Scale bar, 2 μm. **(B)** Graph shows Pearson's and Manders' colocalization coefficients. The data are presented as the mean ± SD. *t* test with equal variances: **$P < 0.001$, n = 5 (five images taken from independent wells with 5–10 cells per image). **(C)** Representative co-IP experiment from control (33) and HD (180) ISPNs treated with bleomycin (10 μg/ml, 2 h) and untreated (NT). Total cell lysates were prepared, and HTT complexes were immunoprecipitated using antibodies to total HTT (MCA2050). TCERG1 and HTT proteins were detected in the IPs. IgG negative control IPs are shown at the bottom panel. The inputs for HTT,

HTT from ISPNs in the co-IP assay (middle panels), suggesting potential interaction with phospho-HTT (Fig 6C).

PLA confirmed proximity (suggesting interaction) of endogenous pS1181-HTT and DNA-PKcs (Fig 6D and E). We observed a dramatic increase in PLA sites concentrating in the nucleus of normal ISPNs after induction of DSBs by bleomycin. (PLA site measurements were reported relative to technical negative single antibody controls. Representative images of negative controls are shown in Fig S3A). Notably, a substantial number of PLA signals were detected in the cytoplasm, especially in untreated cells. Recently, DNA-PKcs has been shown to localize not only in the nucleus but also in the cytoplasm, phosphorylating various substrates involved in cellular metabolism and cytokine production. In addition to its primary function in NHEJ, DNA-PKcs performs other functions in the cytoplasm such as acting as a sensor for double-stranded DNA (dsDNA) to trigger innate immune responses and regulation of glycolysis (68, 69, 70, 71). We have also observed some cytoplasmic localization of DNA-PKcs in our ISPNs (examples are shown in Fig S3B). Significantly less PLA signals were detected in HD cells with a decrease in the number and intensity in the nucleus upon bleomycin treatment. These observations are consistent with disrupted interactions between DNA-PKcs and HTT in HD cells with potential implications for DSB repair processes.

DNA-PKcs is autophosphorylated at multiple sites (72). Phosphorylation at S2056 (within the PRQ cluster) is induced by DSB (73) but is not required for DNA-PK activation (74, 75). We observed an increase in DNA-PKcs phosphorylation (S2056) in normal ISPNs (33CAG) upon induction of DSB with bleomycin (Fig 7A). The levels of pDNA-PKcs were dramatically decreased in HD ISPNs (180CAG) with no induction by DSB, suggesting dysregulation of at least some processes within DSB repair pathways.

pDNA-PKcs was previously found to accumulate at nuclear speckles (NS) outside of DNA damage sites at late time points upon DSB induction suggesting a role of DNA-PK in the regulation of alternative splicing during genotoxic stress (76). In our ISPN model, we also observed accumulation of pDNA-PKcs at SC35-positive nuclear speckles 2 h upon induction of DSB (Fig 7B). Mean Manders' colocalization coefficient reflecting co-occurrence of pDNA-PKcs and SC35 channels was 0.94, whereas mean Pearson's coefficient (which measures the correlation of intensities between two channels) was 0.83, suggesting a high degree of colocalization, though no significant difference between normal (33CAG) and HD (180CAG) neurons was observed (Fig 7C). This colocalization is particularly intriguing because we also found colocalization of phospho-HTT (pS1181) with NS (Fig 3) suggesting potential HTT/DNA-PK interaction within NS.

## Huntingtin (HTT) interacts with BAF chromatin remodeling complex and with Mediator complex

Transcription and chromatin remodeling were the most prominent highly enriched categories among the proteins that increased in abundance in HD ISPNs (based on FC > 1.2), which may suggest a stronger interaction with expanded HTT. Metascape PPI enrichment analysis identified two protein complexes of particular interest (Fig 8A): the first comprises most of the components of BAF chromatin remodeling complex, and the second includes five subunits of transcriptional Mediator complex, including MED15, whose locus is a genetic modifier for HD found in the recent GWAS (47). We aimed to further characterize HTT interactions with components of BAF chromatin remodeling complex closely related to DNA repair.

To confirm endogenous interaction of HTT with BAF complex identified in MS experiments, we conducted co-IPs in our ISPN model (Fig 8B). Total cell lysates were prepared from control (33CAG) and HD (180CAG) ISPNs treated with bleomycin (10 μg/ml, 2 h) and untreated (NT), and HTT complexes were immunoprecipitated using antibodies to total HTT (MCA2050). We targeted two components of the BAF complex—BRG1 (SMARCA4) and ARID1A, which were detected in the IPs with specific antibodies.

IF with confocal microscopy showed partial colocalization of HTT with BRG1 and ARID1A (Figs 9A and B and 10A and B). Mean Manders' colocalization coefficient reflecting co-occurrence of BRG1 or ARID1A with HTT was greater than 0.9 suggesting a high degree of colocalization. Mean Pearson's coefficient (which measures the correlation of intensities between two channels) was 0.66 and 0.52 in control and HD cells, respectively, for HTT-BRG1, and 0.8 (control) and 0.69 (HD) for HTT-ARID1A, which may indicate more direct interaction of HTT with ARID1A, while BRG1 being pulled down in IP as a part of the larger complex. Notably, Pearson's colocalization coefficients of HTT with components of the BAF complex were significantly lower in HD (180CAG) neurons compared with control cells. Endogenous interaction of HTT with BRG1 and ARID1A is suggested by PLA (Figs 9C and D and 10C and D). We observed robust PLA signals in the nuclei of both untreated and bleomycin-treated normal (33CAG) ISPNs. (PLA site measurements were reported relative to technical negative single antibody controls. Representative images of negative controls are shown in Fig S3A). A dramatic decrease in PLA site number and intensity was recorded in HD ISPNs with some induction upon bleomycin treatment for BRG1/HTT PLA signals. These observations are consistent with disrupted functional interactions between BAF complex and nuclear HTT in HD cells.

---

TCERG1, and beta-tubulin are shown (right panels; the same lysates were reused for HTT/BRG1/ARID1A co-IPs shown in Fig 8B). **(D)** Proximity ligation assay (PLA) in normal (33CAG) and HD (180CAG) ISPNs treated with bleomycin (10 μg/ml, 2 h) and untreated using TCERG1 and HTT antibodies. Images shown for 180CAG ISPNs were taken at higher intensity for illustrative purpose, whereas quantitation was done with the same settings as for 33CAG ISPNs. Scale bar, 10 μm. **(E)** Graphs show quantification of PLA signals using MetaXpress software (Molecular Devices). The data are presented as the mean ± SEM of the number of nuclear PLA sites per cell, sum intensity of nuclear PLA sites per cell, and average nuclear PLA site intensity relative to technical negative control within each experiment. Y-axis labels are displayed at the top of each graph. One-way ANOVA with pairwise multiple comparison procedures (Holm–Sidak's method): $*P < 0.01$, $**P = 0.001$, n = 3 (three images taken from independent wells, with 15–20 cells per image). $t$ test with equal variances: $^{\wedge}P = 0.005$, $^{\wedge \wedge}P < 0.001$, $^{\#}P = 0.016$, $^{\#\#}P = 0.003$, n = 3 (three images taken from independent wells with 15–20 cells per image).

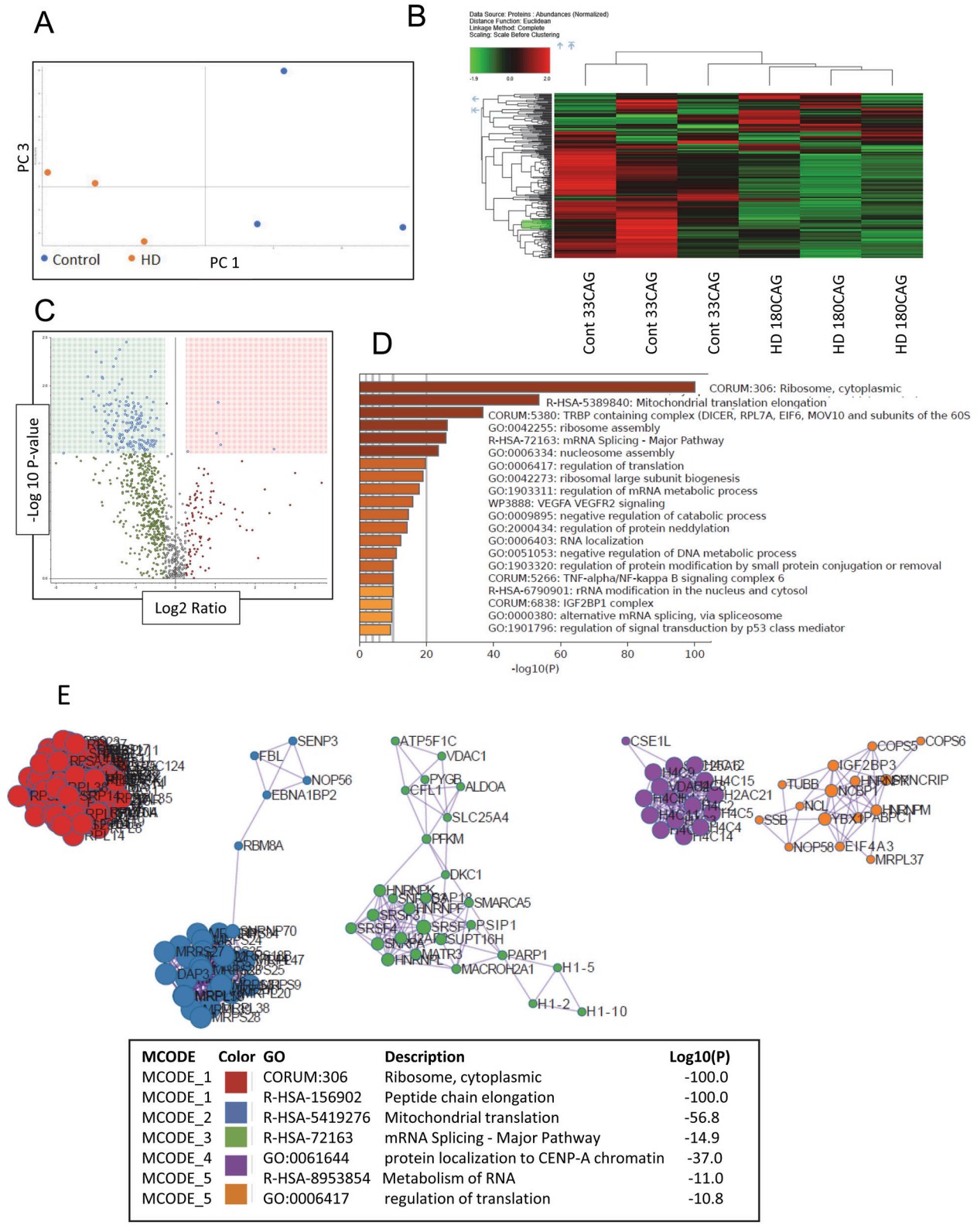

**Figure 5. Unbiased screen for HTT interactors upon induction of DSB/DDR.**
**(A, B, C)** PCA plots, (B) heat map, and (C) volcano plot comparing protein abundance between HD and control samples after normalizing to HTT abundance (PD2.4). Green and pink boxed areas represent 20% fold change (FC) at $P < 0.05$. **(D)** Pathway and process enrichment analysis of proteins decreased (HD/control < 0.8) in abundance in HTT pulldowns from HD ISPNs relative to control ISPNs (generated using "Metascape" analysis tools) (65). "Log$_{10}$(P)" is the enrichment $P$-value in log base 10. **(E)** Protein–protein interaction (PPI) enrichment analysis generated by "Metascape" of proteins decreased in abundance in HD ISPNs versus control neurons. The table shows best-scoring terms by $P$-value with functional description of the corresponding MCODE components.

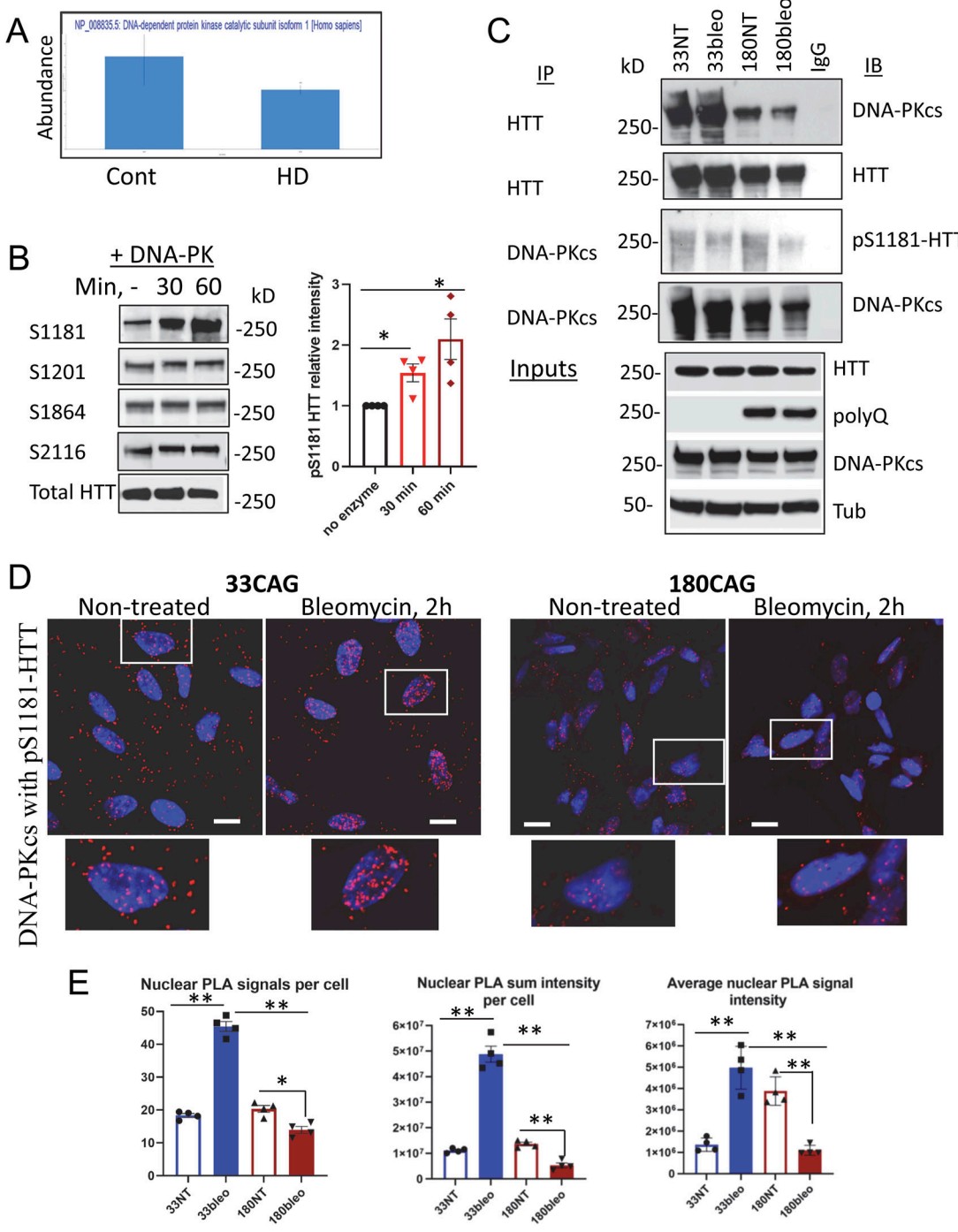

**Figure 6. HTT is phosphorylated by and interacts with DNA-PKcs.**
**(A)** Quantitative channel value view (exported from PD2.41) for DNA-PKcs (PRKDC) showing grouped abundance from the pulldowns of control and HD ISPNs. **(B)** In vitro phosphorylation assay with the recombinant HTT-23Q/HAP40 complex (Curia). The DNA-PK complex (Promega), activated by the addition of linear double-stranded DNA, was incubated with HTT/HAP40 for indicated times, and phosphorylation state of HTT was assessed using indicated phospho-specific antibodies. The graph shows quantitation of phospho-HTT normalized to total HTT. Data represent the mean ± SEM. Mann–Whitney rank sum test: *$P$ = 0.029, n = 4. **(C)** Representative co-IP experiments from control (33CAG) and HD (180CAG) ISPNs treated with bleomycin (10 µg/ml, 2 h) and untreated (NT). Total cell lysates were prepared, and HTT complexes were immunoprecipitated using antibodies to total HTT (MCA2050, two top panels). DNA-PKcs and HTT were detected in the immunoprecipitates (IPs). In reverse, DNA-PKcs protein was immunoprecipitated (two middle panels) and phospho-HTT protein was detected in the eluates using a specific antibody to pS1181. The inputs for HTT, DNA-PKcs, and beta-tubulin are shown below the IPs. **(D)** Proximity ligation assay (PLA) of normal (33CAG) and HD (180CAG) ISPNs treated with bleomycin (10 µg/ml, 2 h) and untreated using pS1181-HTT and DNA-PKcs antibodies. Scale bar, 10 µm. **(E)** Graphs show quantification of PLA signals using MetaXpress software (Molecular Devices). The data are presented as the mean ± SEM of the number of nuclear PLA sites per cell, sum intensity of nuclear PLA sites per cell, and average nuclear PLA site intensity relative to technical negative control within each experiment. Y-axis labels are displayed at the top of each graph. One-way ANOVA with pairwise multiple comparison procedures (Holm–Sidak's method): **$P$ < 0.001, *$P$ = 0.004, n = 4 (4 images taken from independent wells with 10-20 cells per image).

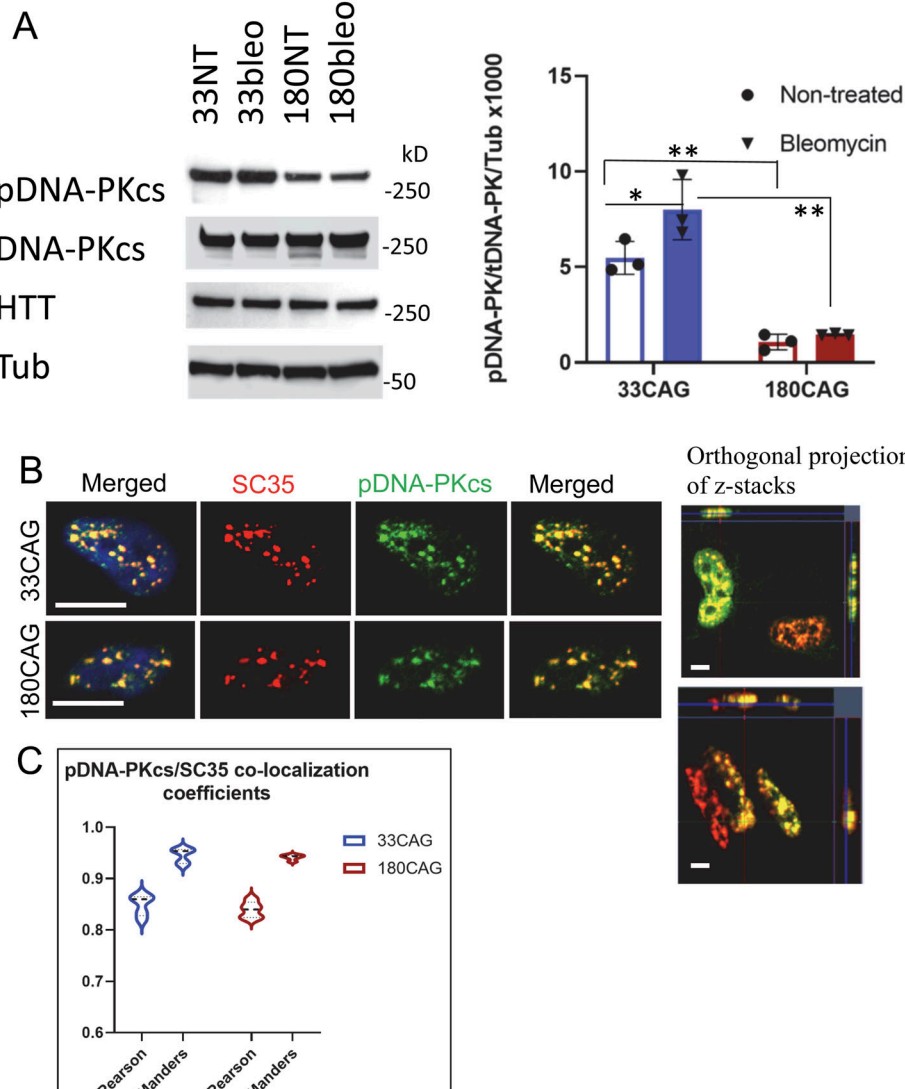

**Figure 7. Phosphorylated DNA-PKcs (S2056) is associated with nuclear speckles and is decreased in HD cells.**
**(A)** Levels of pDNA-PKcs (S2056) are decreased in HD ISPNs. Western blot with indicated antibodies of total cell lysates prepared from control (33) and HD (180) ISPNs treated with bleomycin (10 µg/ml, 2 h) and untreated (NT). Graph shows quantitation of phospho-DNA-PKcs normalized to total DNA-PKcs. Data represent the mean ± SEM. Mann–Whitney rank sum test: *$P$ = 0.02, n = 3; **$P$ < 0.001. **(B)** Representative images of ISPNs treated with bleomycin (10 µg/ml, 2 h) costained with pDNA-PKcs (S2056) and SC35 antibodies. Scale bar, 10 µm. Orthogonal projections of z-stacks on right panels demonstrate colocalization. Scale bar, 5 µm. **(C)** Graph shows Pearson's and Manders' colocalization coefficients. The data are presented as the mean ± SD. No significant difference between normal and HD ISPNs was observed ($t$ test with equal variances). n = 5 (five images taken from independent wells with 10–15 cells per image; cells were analyzed individually, and means of coefficients were calculated for each image).

## Ascorbate peroxidase (APEX2)–mediated proximity-based proteomics for HTT protein interactions

Most of the previous HTT interactome studies have used HTT affinity purifications (HTT pulldowns) followed by mass spectrometry. However, such co-IP experiments performed on cell lysates after disruption of cellular structures may capture complexes formed in solution in addition to endogenous interactions. These methods are highly dependent on the antibodies to the bait protein, which may introduce additional bias. Proximity-based proteomics provides several advantages compared with traditional approaches including yeast two-hybrid systems and affinity purifications. Ascorbate peroxidase (APEX2)–based proximity labeling (with APEX2 tagging) can detect weak or dynamic interactions and can preserve the intracellular spatial information of the interactomes. The APEX2 enzyme mediates biotinylation of proteins located in its proximity in living cells with intact membranes and protein complexes, and this method has a potential to detect the temporal changes of the HTT interactome in response to acute cellular stresses. TMT-mediated proteomics workflow can be performed after enrichment for biotinylated peptides to quantify potential interactors. To our knowledge, only one HTT interactome study using proximity proteomics has been reported (77).

To test the APEX2-labeling approach, we have generated APEX2-fusion constructs with full-length WT (23Q) and polyQ-expanded (82Q) HTT, and with two HTT-82Q variants with altered PTM sites (S116A and S1181A). The plasmids were transfected into HEK293 cells with four samples analyzed for each group (plus two samples for negative control without biotin). Biotin labeling was performed as described previously (78), and conditions were optimized to ensure sufficient biotinylation, enrichment, and reduction of nonspecific binding (Fig S4). No biotin (lane 2) was

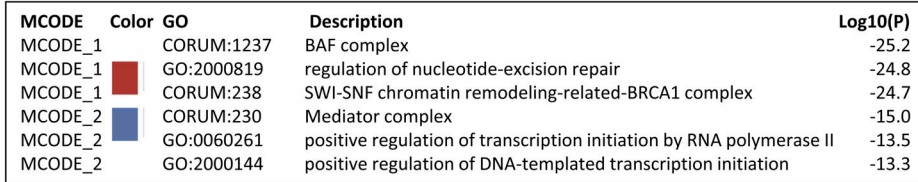

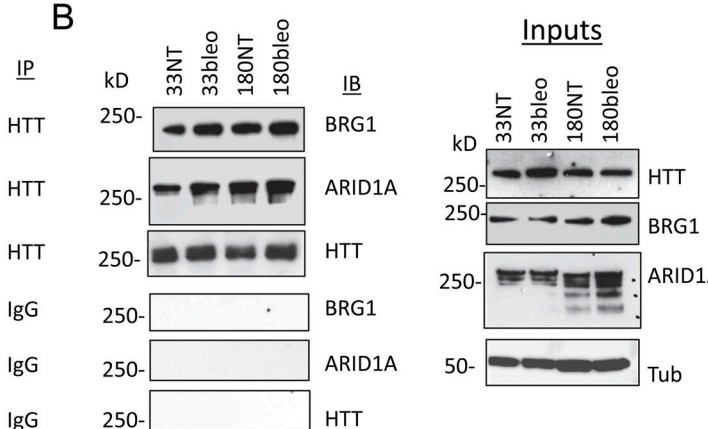

**Figure 8. Huntingtin (HTT) interacts with components of the BAF chromatin remodeling complex.**
**(A)** BAF chromatin remodeling and Mediator complexes identified among HTT-associated proteins (Metascape analysis). **(B)** Representative co-IP experiments from control (33) and HD (180) ISPNs treated with bleomycin (10 μg/ml, 2 h) and untreated (NT). Total cell lysates were prepared, and HTT complexes were immunoprecipitated using antibodies to total HTT (MCA2050). BRG1, ARID1A, and HTT were detected in the immunoprecipitates (IPs). IgG negative control IPs are shown at the bottom panels. The inputs for HTT, BRG1, ARID1A, and beta-tubulin are shown (right panels; the same lysates were reused for HTT/TCERG1 co-IPs shown in Fig 4C).

| MCODE | Color | GO | Description | Log10(P) |
|---|---|---|---|---|
| MCODE_1 | | CORUM:1237 | BAF complex | -25.2 |
| MCODE_1 | (red) | GO:2000819 | regulation of nucleotide-excision repair | -24.8 |
| MCODE_1 | | CORUM:238 | SWI-SNF chromatin remodeling-related-BRCA1 complex | -24.7 |
| MCODE_2 | | CORUM:230 | Mediator complex | -15.0 |
| MCODE_2 | (blue) | GO:0060261 | positive regulation of transcription initiation by RNA polymerase II | -13.5 |
| MCODE_2 | | GO:2000144 | positive regulation of DNA-templated transcription initiation | -13.3 |

added for negative control. Biotinylated proteins were enriched by streptavidin pulldown and detected in the inputs and in the eluted samples (but not in negative controls). Fewer biotinylated proteins in flow-through indicate efficient binding to streptavidin resin.

To increase the specificity of the assay, we have modified previously published protocol (77, 78) and performed trypsin digestion of the lysates first, followed by enrichment for biotinylated peptides, instead of a pulldown of biotinylated proteins (which may include proteins nonspecifically binding to streptavidin resin) (see the Materials and Methods section for details). 82% (1,315 out of 1,603) of peptide groups identified using this approach contained biotinyl-tyramide modification, representing 350 proteins identified with high confidence (1% FDR), and quantified (Table S2; only data for unmodified HTT-23Q and HTT-82Q are shown). The raw protein abundance was normalized to the APEX2 protein (within Protein Discoverer workflow) and used to calculate the protein-level ratios to negative control (no biotin added). To ensure specificity, we focused on 242 proteins, found in all samples, which showed sample/negative control ratios of at least 10. This set of proteins found in proximity of HTT (and thus potential HTT interactors) was selected for pathway and process enrichment analysis using "Metascape" tools (65).

The most enriched GO terms in this dataset were metabolism of RNA and regulation of translation, as well as protein folding and

stress response (Fig 11A). Protein–protein interaction (PPI) enrichment analysis (Fig 11B, Metascape) identified networks with the highest enrichment P-values: metabolism and transport of RNA, translation, protein localization to chromatin, and protein folding. Thus, although preliminary, this analysis highlights the same functional categories as identified using the co-IP-MS approach (above).

Using this approach in transfected HEK293 cells, we found minimal differences in the interactomes of WT (23Q) and expanded (82Q) HTT variants. We detected 16 HTT peptides with biotinyl-tyramide modifications, which may be a result of intramolecular proximity of the N-terminal APEX2 tag on HTT to its other domains. In addition, endogenous WT HTT may come close and form complexes with transfected APEX2-tagged HTT-82Q, thus producing a background of biotinylated proteins that are proximal to both exogenous transfected and endogenous WT HTT. Because "biotinylation radius" is not a fixed value, the endogenous proteins that are many nanometers away from APEX2 may still be biotinylated, although weakly, and detected by MS (78). This background may account for minimal differences between APEX2-tagged WT and expanded HTT interactomes. To overcome this difficulty, it would be beneficial to engineer APEX2 tag on the endogenous HTT (e.g., by CRISPR) in the future studies. The APEX2 tag can also be placed at the C terminus or

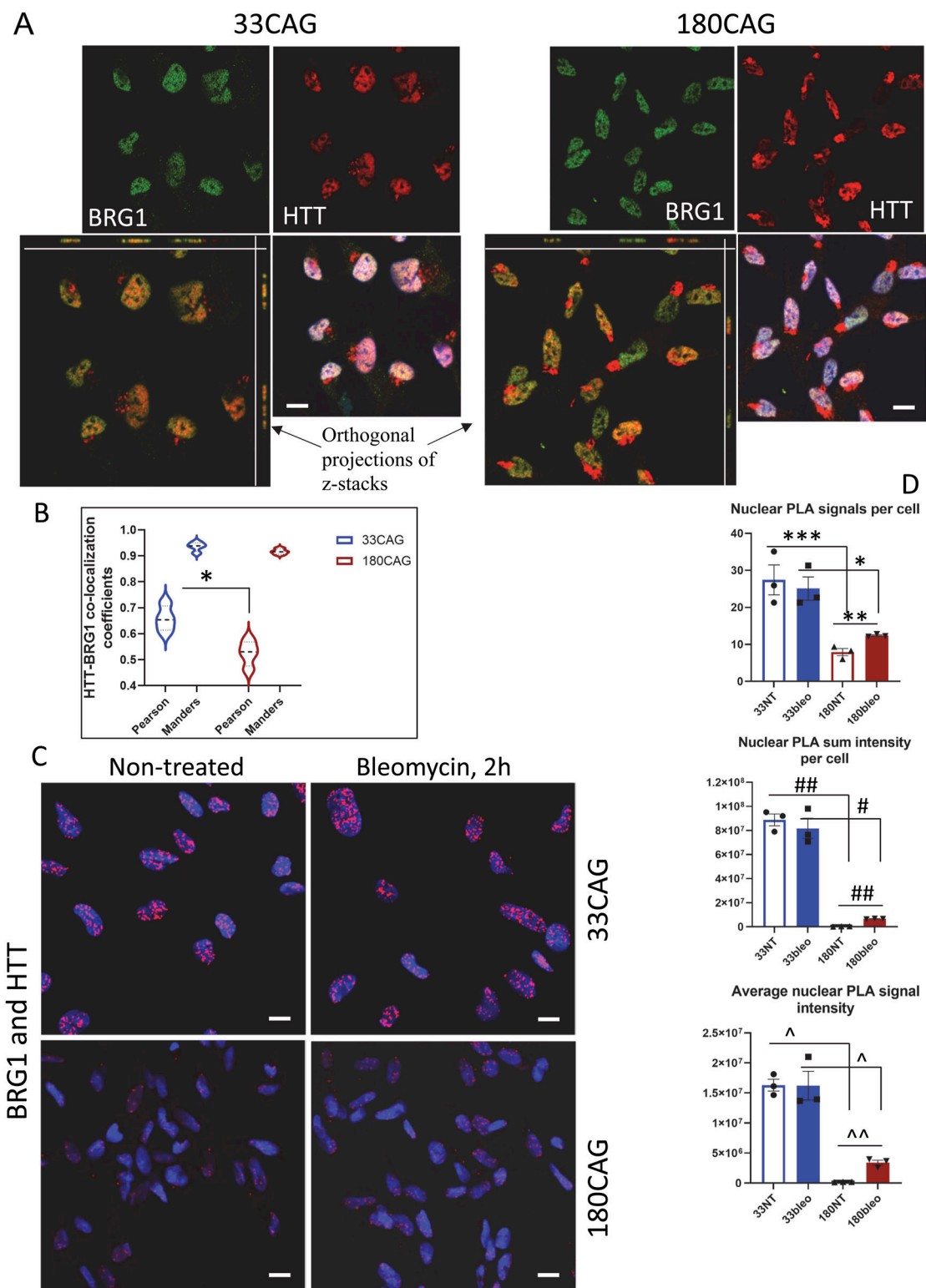

**Figure 9. HTT/BRG1 (SMARCA4) colocalization and direct interaction in ISPNs.**
**(A)** Representative images of ISPNs treated with bleomycin (10 μg/ml, 2 h) costained with BRG1 and HTT (MCA2050) antibodies. Orthogonal projections of Z-stacks (shown for red and green merged panels) demonstrate colocalization. Scale bar, 10 μm. **(B)** Graph shows Pearson's and Manders' colocalization coefficients. The data are presented as the mean ± SD. *t* test with equal variances: *$P$ = 0.009, n = 4 (four images taken from independent wells with 15–20 cells per image). **(C)** Proximity ligation assay (PLA) in normal (33CAG) and HD (180CAG) ISPNs treated with bleomycin (10 μg/ml, 2 h) and untreated using BRG1 and HTT antibodies. Images shown for 180CAG ISPNs were taken at higher intensity for illustrative purpose, whereas quantitation was done with the same settings as for 33CAG ISPNs. Scale bar, 10 μm. **(D)** Graphs show quantification of PLA signals using MetaXpress software (Molecular Devices). The data are presented as the mean ± SEM of the number of nuclear PLA sites per cell, sum intensity of nuclear PLA sites per cell, and average nuclear PLA site intensity relative to technical negative control within each experiment. Y-axis labels are displayed at

within other portions of HTT to further explore domain-specific interactions.

## Discussion

We explored HTT protein interactions using multiple approaches including novel proximity-based methods, which can preserve the intracellular spatial information of the interactomes. Our analyses support the role of HTT in the regulation of interconnected DNA repair/remodeling, RNA processing, and protein translation pathways.

Previous HTT interactome studies (10, 11, 12) revealed HTT involvement with other cytoplasmic processes such as actin cytoskeleton regulation, axonal transport vesicular trafficking, mitochondrial dysfunction, and chaperone-mediated stress response. Recent data (79) highlight the role of HTT in protein translation beyond translation of mHTT itself, and that accumulating mHTT protein fragments sequester eIF5A leading to progressive, age-dependent depletion of eIF5A in HD cells and mouse brains promoting widespread ribosome pausing and altered translation elongation kinetics. Our data presented here, as well as other current findings, highlight nuclear mechanisms such as chromatin remodeling, transcription, and splicing. Thus, HTT may be in a central position for regulation of multiple cellular accommodations to DNA damage stress.

Although we find that HTT interacts with several proteins involved with DNA repair, it is notable that there appears to be little or no interaction with the proteins encoded by genetic modifier loci (17, 18, 19, 20, 21, 22, 23, 80), many of which are specifically involved in mismatch repair. We do find interactions with MED15 and TCERG1, both of which are encoded by genetic modifier loci. Two other loci, *RRM2B* and *CCDC82*, encode for proteins whose function is not well understood. RRMB2 is induced by *p*53 and appears to be involved in mitochondrial DNA maintenance (81). CCDC82 undergoes ATM-dependent phosphorylation after H2O2 exposure (82). Thus, DNA repair and maintenance appear to be common themes in HD pathogenesis. The interaction of full-length mHTT with products of genetic modifier genes such as MED15 and TCERG1 could provide indirect evidence for a role of full-length mHTT (rather than exon 1 or other mechanisms) in at least some aspects of HD pathogenesis.

Phospho-HTT is implicated in protein interactions described in this study. There is solid support in the literature for the role of PTMs in HTT functionality and perhaps in mutant HTT toxicity (6, 54). Enzymes catalyzing some of these modifications have been suggested as potential therapeutic targets (51, 52, 53, 55, 56, 83). We have previously identified several dozen phosphorylation, acetylation, and arginine methylation/dimethylation sites from HD knock-in mice, human postmortem brain, and immortalized striatal precursor neurons, and defined their effects on HTT solubility and toxicity (14, 52, 57).

We show here that S1181 phosphorylation is regulated by DSB and can be carried out (at least in vitro) by a major DSB kinase DNA-PK. This is particularly intriguing because we found both phospho-HTT and DNA-PKcs proteins colocalizing with NS. Association of HTT with SC35-positive speckles was first noticed two decades ago (84). HTT phosphorylated at the N-terminal sites was localized to SC35-positive speckles upon oxidative stress (85). In another recent study (86), phospho-N17-HTT was found to colocalize with both SC35-positive nuclear speckles and NEAT1 paraspeckles. In our current study, we found that HTT phosphorylated at the previously characterized S1181 site (51, 52, 57, 83) forms nuclear puncta colocalizing with the NS marker SC35 (SRSF2).

As previously suggested (76), pDNA-PKcs localized to NS may participate in the regulation of alternative splicing during genotoxic stress. HTT/DNA-PKcs interaction and localization of both pDNA-PKcs and pS1181-HTT proteins to NS are indicative of potential functional cooperation in regulation of DNA repair and RNA processing, although the exact mechanism and physiological outcome of HTT interaction with DNA-PKcs shown here remain to be examined. Liu et al (76) found that DNA-PK inactivation affects splicing of a set of pre-mRNAs, including major splicing regulators SRSF1 and SRSF2. Notably, both splicing factors have been shown to be mis-spliced in HD (31, 33).

Autophosphorylation of DNA-PKcs at the PQR cluster (which includes S2056) participates in DNA end processing and the pathway choice for DSB repair (74, 87). Several studies reported that phosphorylation at this site is not required for activation of DNA-PKcs, and it has been suggested to facilitate the dissociation of DNA-PKcs from DNA ends at damage sites (74, 75, 88). Considering that S2056 can be phosphorylated in response to ionizing radiation that induces DSB (73), it can be used as a measure of an active DNA repair, or a completion of such and dissociation of DNA-PKcs from the complex. We found a dramatic decrease of the pDNA-PKcs protein level in HD ISPNs with no induction by DSB, suggestive of dysregulation of DSB repair processes.

Although NS have been primarily associated with RNA splicing and storing of multiple splicing factors, they also harbor proteins crucial for epigenetic regulation, chromatin organization, transcription, and DNA repair (59). Furthermore, an extensive biochemical study (89) demonstrated a stable association of NS with chromatin, and spatial proximity to NS was shown to directly correlate with amplification of gene expression (90). In this model, transcriptional activation is boosted by NS, which deliver RNA processing factors to the nascent RNAs. Thus, NS may play a direct role in facilitating integrated regulation of gene expression (reviewed in references 91, 92).

We have previously found HTT interactions with a few RBPs, also found in NS, including FUS, EWSR1, and HNRNPUL1 (35). Transcription-related factors—TCERG1—and components of the Mediator complex were also found associated with NS (64, 93). A recent study (94) suggests a link between the Mediator complex and NS, with Med15, a subunit in the tail module of the Mediator complex, forming nuclear condensates. BAF complex subunits

---

the top of each graph. *t* test with equal variances: \*$P$ = 0.016, \*\*$P$ = 0.011, \*\*\*$P$ = 0.009, #$P$ = 0.005, ##$P$ < 0.001, ^$P$ = 0.006, ^ ^$P$ = 0.002, n = 3 (three images taken from independent wells with 10–20 cells per image).

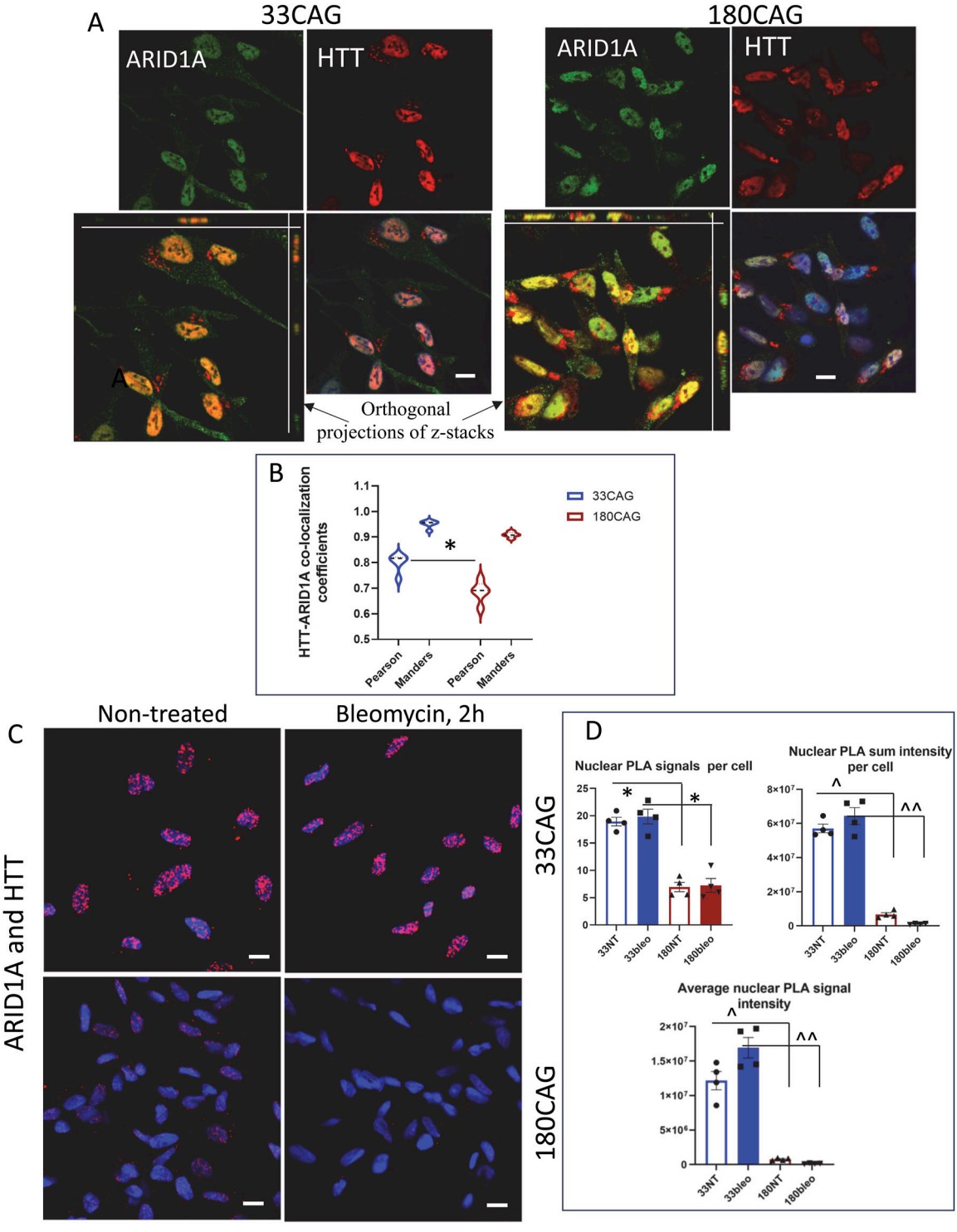

**Figure 10. HTT/ARID1A colocalization and direct interaction in ISPNs.**
**(A)** Representative images of ISPNs treated with bleomycin (10 $\mu g$/ml, 2 h) costained with ARID1A and HTT (MCA2050) antibodies. Orthogonal projections of Z-stacks (shown for red and green merged panels) demonstrate colocalization. Scale bar, 10 $\mu m$. **(B)** Graph shows Pearson's and Manders' colocalization coefficients. The data are presented as the mean ± SD. $t$ test with equal variances: *$P$ = 0.002, n = 5 (five images taken from independent wells with 10–20 cells per image). **(C)** Proximity ligation assay (PLA) in normal (33CAG) and HD (180CAG) ISPNs treated with bleomycin (10 $\mu g$/ml, 2 h) and untreated using ARID1A and HTT antibodies. Images shown for 180CAG ISPNs were taken at higher intensity for illustrative purpose, whereas quantitation was done with the same settings as for 33CAG ISPNs. Scale bar, 10 $\mu m$. **(D)** Graphs show quantification of PLA signals using MetaXpress software (Molecular Devices). The data are presented as the mean ± SEM of the number of nuclear PLA sites per cell, sum intensity of nuclear PLA sites per cell, and average nuclear PLA site intensity relative to technical negative control within each experiment. *One-way ANOVA with pairwise multiple comparison procedures (Holm–Sidak's method): $P$ < 0.001, ^$t$ test with equal variances. $P$ < 0.001, ^ ^Mann–Whitney rank sum test. $P$ = 0.029, n = 4 (four images taken from independent wells with 10–20 cells per image).

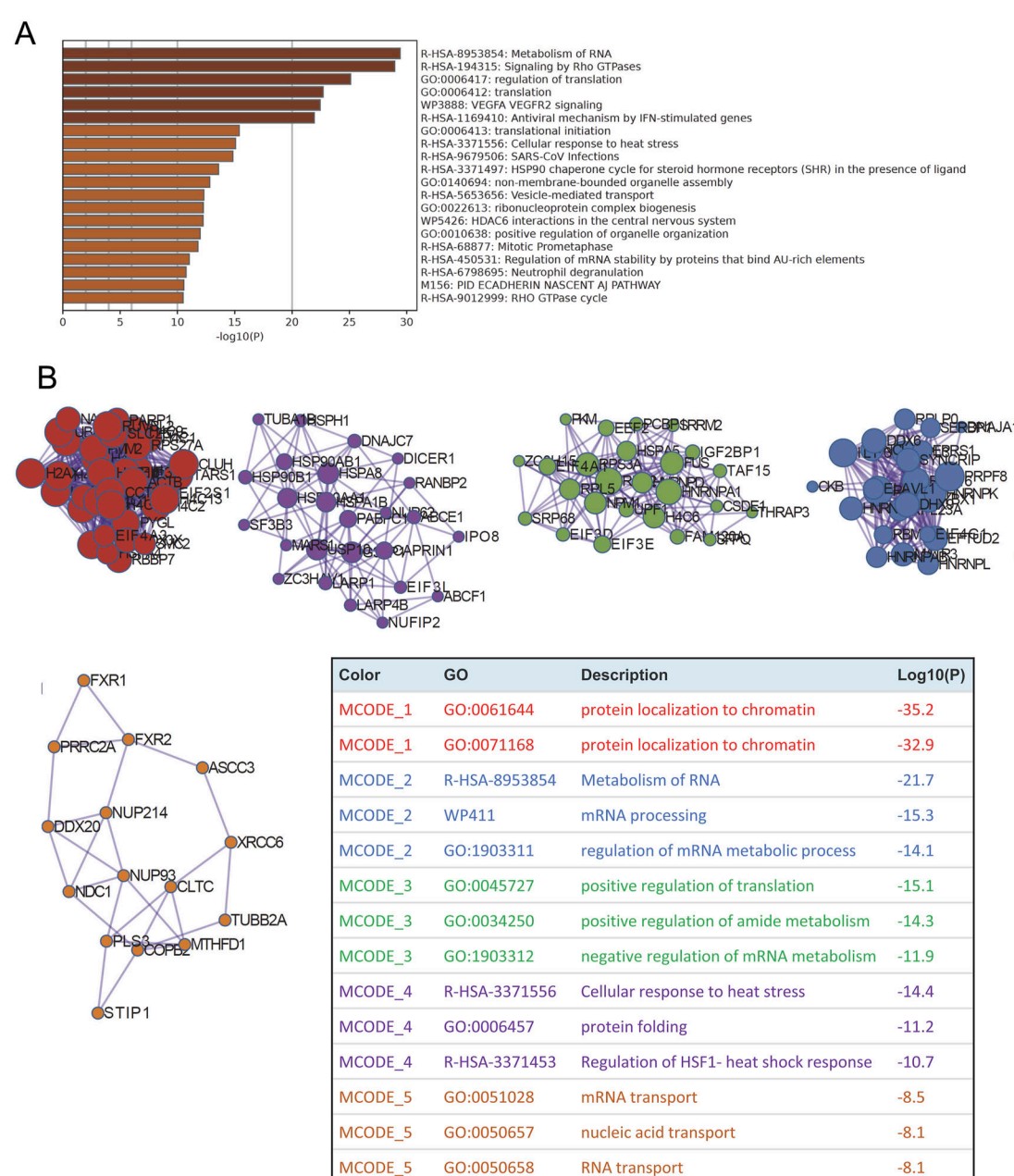

**Figure 11. APEX2-mediated proximity labeling for HTT interactions.**
**(A)** Pathway and process enrichment analyses of proteins found in proximity of HTT with at least one biotinylated (biotinyl-tyramide) peptide (generated using "Metascape" analysis tools, reference (65)). "Log$_{10}$(P)" is the enrichment *P*-value in log base 10. **(A, B)** Protein–protein interaction (PPI) enrichment analysis generated by "Metascape" of the same database as in (A). The table shows best-scoring terms by *P*-value with functional description of the corresponding MCODE components.

have been shown to interact with splicing factors or other proteins localized to NS, such as RNA polymerase II (95), suggesting a mechanism where chromatin remodeling at specific genes could influence RNA splicing. Two other current studies (29 *Preprint*, 96) report HTT interaction with both BAF and Mediator complexes in the mouse brain and SH-SY5Y cell model. Thus, the interaction of HTT with TCERG1, Mediator complex, and chromatin remodelers, which we found in this study, may occur at least in part in NS, facilitating communication between transcription and RNA splicing.

In our co-IP/mass spectrometry study, we found higher abundance of the subunits of the BAF complex, BRG1 and ARID1A, in HTT pulldowns from HD ISPNs, which may indicate increased interaction. However, using a proximity-based method (PLA), we found a reduction in the nuclear PLA signals between HTT and subunits of the BAF complex in HD ISPNs. This discrepancy may reflect different methodologies used in the above two approaches: in the co-IP approach, HTT was pulled down from the total cell lysate using a mix of MCA2050/2051 antibodies that detect both nuclear and cytoplasmic HTT, so this method does not account for potential differences in

cytoplasmic/nuclear distribution of HTT in normal and HD neurons. Notably, proteomics quantitation data are normalized to the amount of HTT, so there is no bias because of different affinities of the antibodies toward wild-type or mutant HTT, or potential differences in HTT levels in control and HD cells. In contrast, in situ proximity-based method (PLA) can preserve the intracellular spatial information of the interactomes and its amplified readout signal is highly dependent on the levels of the protein of interest in specific subcellular locations and, consequently, the number of antibody molecules bound to the target. However, the potential contribution of different affinities of the antibodies toward wild-type or mutant HTT cannot be excluded.

We suggest that depletion of nuclear HTT in HD neurons may partially account for decreased nuclear PLA signals for BRG1/HTT and ARID1A/HTT proximity. In support of this hypothesis, we previously found (using 3D quantification with Imaris) lower HTT immunoreactivity in the nucleus of HD ISPNs relative to normal control cells (14). As illustrated in Fig S5, we observed the decreased nuclear presence of HTT in most HD ISPNs and increased HTT localization to perinuclear structures positive for the Golgi marker, GM130.

Notably, we observed some cytoplasmic PLA sites for DNA-PKcs/HTT and ARID1A/HTT proximities especially evident in HD ISPNs (Figs 6D and 10C). This shift toward cytoplasmic proximity in combination with depleted nuclear HTT may in part account for decreased nuclear PLA signals in HD neurons. Cytosolic interactions of typically nuclear DNA repair/remodeling proteins with the mutant HTT could impact the function of these complexes, thus exacerbating transcriptional abnormalities observed in HD.

Proteins containing low complexity regions (LCRs), generally corresponding to intrinsically disordered regions (IDRs), are overrepresented among NS proteins. Such proteins, including HTT (14, 97), are prone to phase separation in the conditions of cellular crowding, which is considered a major force maintaining the integrity of NS lacking lipid membranes. Several studies demonstrated NS morphology changes after transcriptional inhibition and genotoxic stress (89, 98). Kim et al (98) confirmed liquid state properties of NS and tracked the motion and fusion of speckles into bigger and increased intensity droplets after transcriptional inhibition or stress responses. This was attributed to splicing factors diffusing back into speckles from active transcription sites. We found that NS appeared disorganized in HD ISPNs with a few large speckles formed in some cells, which would be consistent with an increased stress burden in HD ISPNs exacerbated by the induction of DNA damage. Decreased viability of HD cells upon ER and oxidative stress has been demonstrated (85, 99). Here, we show an increased vulnerability of HD patient-derived neurons to genotoxic stress including reduced viability (ATP) and increased apoptosis (activation of caspase 3/7) after induction of DSB by bleomycin.

Many splicing factors found in NS are modified posttranslationally, and these PTMs within their LCRs may regulate their protein interactions, function, and association with NS. We showed previously that HTT interacts with PRMT (protein arginine methyltransferase) enzymes and affects their activity (14). Arginine methylation, a modification that plays a major role in the function of RBPs (reviewed in references 100, 101), was decreased for several NS proteins in HD neurons (35). Here, we found that HTT can also interact with DNA-PKcs, a major DDR kinase that has numerous

substrates not only among DNA repair proteins, but also among RBPs, such as FUS, which is phosphorylated by DNA-PK after DNA damage and binds DSB (37). Thus, the mutant HTT may alter RNA processing and transcription via abnormal interactions with modifying enzymes (PRMTs, kinases) and other DDR proteins involved in the maintenance of genome stability.

In conclusion, our study highlights the role of HTT in the emerging integration and interplay between DNA repair/remodeling and RNA processing pathways (reviewed in references 40, 102, 103) mediated in part by DDR kinases regulating RNA metabolism in response to DNA damage. The activation of the DDR can modify RNA splicing by affecting expression and modifications of various transcription, splicing, and mRNA export proteins (102). Several DDR proteins and RBPs have a dual function in RNA processing and DNA repair and may be directly involved in the prevention and repair of DNA lesions (40). RNA may play a direct role in facilitating DNA repair: it was shown recently that NHEJ proteins form a complex with RNA polymerase II and can use nascent RNA as a template for error-free DSB repair of transcribed genes (104, 105).

Chromatin remodeling is also closely related to DNA repair: DDR kinases regulate the chromatin and nucleosome rearrangements near a DSB to provide a scaffold for the recruitment of other DDR factors and inhibit local transcription. DDR-dependent H2A.X phosphorylation induces changes to chromatin structure by recruiting ATP-dependent chromatin remodeling complexes including SWI/SNF (48). Thus, genotoxic stress affects chromatin remodeling and RNA splicing activity via modification and reengagement of splicing factors and other mechanisms, whereas nuclear speckles emerge as multifunctional organelles providing an integrated regulation of gene expression and RNA processing.

We suggest that HTT functional interactions with modifying enzymes (such as DDR kinase DNA-PK and PRMTs), and its association with NS, with numerous RBPs, and with chromatin remodeler BAF may position HTT as a unique scaffolding intermediary providing interconnections between DNA repair/remodeling and RNA processing pathways. Further characterization of the functional consequences of these interactions will help elucidate the impact of HTT-lowering therapy and may provide important insights into HD pathogenesis and therapeutic targets.

# Materials and Methods

### Cells

Immortalized striatal precursor neurons (ISPNs) were generated in our laboratory from patient-derived iPS cells with normal (33) or expanded (180) CAG repeats using co-expression of the enzymatic component of telomerase hTERT and drug-regulated c-Myc as described previously (46). HD and nondisease repeat iPSCs were generated and characterized as described previously (8, 13) from human fibroblast cell lines obtained from the Coriell Institute for Medical Research, under their consent

and privacy guidelines as described on their website (http://catalog.coriell.org/). The use of human materials was approved and registered with the Johns Hopkins Biosafety Office and Institutional Biosafety Committee, Registration No. B9210230332. Notably, our 180CAG HD ISPN line has been undergoing CAG repeat expansion and the CAG size at the time of these studies ranged from 240 to 260. For clarity, we are referring to this line as 180CAG throughout this study. The ISPNs were propagated as stable adherent neuronal precursors using Matrigel-coated plates with SCM2 medium (46) at 37°C and 5% $CO_2$. Differentiation to a phenotype resembling medium spiny neurons was performed as previously described (46). Human embryonic kidney (HEK) 293FT cells were from Invitrogen (Thermo Fisher Scientific) and were grown in DMEM (with 4.5 g/liter D-glucose; Thermo Fisher Scientific) supplemented with 10% FBS, 100 $\mu$g/ml geneticin, 100 units/ml penicillin, and 100 units/ml streptomycin. For induction of DSBs, cells were treated with bleomycin (MilliporeSigma) at 10 $\mu$g/ml for 30 min or 2 h as indicated.

### Cell viability/apoptosis assays

For cell viability/apoptosis assays, $1 \times 10^4$ ISPN cells/well were plated in Matrigel-coated 96-well plates. 24 h after plating, adenosine triphosphate (ATP) levels were measured using CellTiter-Glo Luminescent Cell Viability Assay (Promega) in triplicate wells for each condition. Each individual well was assumed as biological replicate for statistical analysis. For cell apoptosis assay, we used Caspase-Glo 3/7 Assay System (Promega).

### HTT in vitro phosphorylation assay

HTT/HAP40 complex purification was conducted by Curia. HAP40 was added as stabilizing binding partner (106). Baculoviruses expressing C-terminal FLAG-tagged HTT-23Q and N-terminal Myc-tagged HAP40 were produced, and HEK293 cells were transduced with both viruses. HTT-HAP40 complexes were isolated using FLAG affinity purification. The following buffers were used: cell lysis buffer—50 mM Tris, pH 8, 300 mM NaCl, 10% glycerol, 5 mM EDTA, protease inhibitors; wash buffer—50 mM Tris, pH 8, 300 mM NaCl, 10% glycerol, 10% glycerol, 0.01% Tween-20; and elution buffer—50 mM Tris, pH 8, 300 mM NaCl, 10% glycerol, 0.5% CHAPS, 0.4 mg/ml FLAG peptide.

In vitro phosphorylation assays were carried out with DNA-PK (#V581A; Promega) according to the manufacturer's protocol. 200 units of the DNA-PK complex, activated by the addition of 10 $\mu$g/ml of calf thymus DNA, was mixed with 5ug of the purified HTT/HAP40 complex for indicated times, and the phosphorylation state of HTT was assessed using phospho-specific antibodies to S1181, S1201, S1864, and S2116 (57).

### IP and Western blotting

For detection of endogenous HTT, the following antibodies were used: MCA2050 (HDB4E10, 1:1,000 dilution; Bio-Rad) directed to amino acids 1,844–2,131 of HTT, MCA2051 (HDC8A4, 1:1,000 dilution; Bio-Rad) directed to amino acids 2,703–2,911 of HTT, and MW1 anti-

polyQ (Developmental Studies Hybridoma Bank, University of Iowa, 1:1,000 dilution) to detect mutant HTT in HD ISPNs. Antibodies to phospho-HTT (S1181, S1201, S1864, S2116) were used at 1:1,000 dilution and have been previously described and validated (see below) (57, 58). Other antibodies used were as follows: ATM (2C1) from Genetex (#GTX70103), 1:1,000 dilution; phospho-ATM (Ser1981) from R&D Systems (#AF1655), 1:1,000 dilution; phospho-histone H2A.X (S139) (3F2) from Genetex (#GTX80694), 1:1,000 dilution; histone H2A.X (D17A3) from Cell Signaling Technology (#7631), 1:1,000 dilution; phospho-DNA-PKcs (S2056) from Abcam (#18192), 1:1,000 dilution; DNA-PKcs (E6U3A) from Cell Signaling Technology (#38168), 1:1,000 dilution; TCERG1/CA150 from Bethyl Laboratories (#A300-360A), 1:1,000 dilution; BRG1 (BLR106H) from Bethyl Laboratories (#A700-106), 1:1,000 dilution; ARID1A/BAF250 (BLR279L) from Bethyl Laboratories (#A700-279), 1:1,000 dilution; and b-actin (C4) from Santa Cruz Biotechnology (#sc-47778), 1:2,000 dilution.

For detection of HTT interactions by co-immunoprecipitation, ISPN cells were collected and lysed in co-IP buffer (50 mM Tris, 150 mM NaCl, 5 mM EDTA, 50 mM $MgCl_2$, 0.5% Triton, 0.5% sodium deoxycholate) in the presence of protease inhibitors (PIC, Protease Inhibitor Cocktail III, Calbiochem) and Halt Phosphatase Inhibitor Cocktail (Thermo Fisher Scientific) followed by centrifugation at 14,000$g$. The resulting supernatants were diluted 1:1 with PBS and were precleared by incubation with protein G agarose for 1 h at 4°C. IPs were carried out ON at 4°C using either antibodies to HTT (1:1 mix of MCA2050 and MCA2051 at 1:200 dilution) or to protein-specific antibodies (described above). The IPs were washed 3 times with the lysis buffer, and protein complexes were eluted from the beads with 2xSDS Laemmli sample buffer (Bio-Rad), fractionated on SDS–PAGE, and detected by Western blotting with antibodies described above. Inputs were collected before IP to control for protein levels.

### Quantitation of the stoichiometry of phosphorylation sites on HTT

Normal or polyQ-expanded mHTT proteins from HTT bands were cut out from the gel after immunoprecipitation from total cell lysates of control (33CAG) and HD (180CAG) ISPNs upon induction of DSB by bleomycin (10 $\mu$g/ml, 30 min), or from untreated cells (NT).

Quantitative targeted parallel reaction monitoring (PRM) mass spectrometry was performed for modified peptides chosen from previous discovery data-dependent acquisitions (DDA) and was carried out as described previously (14). The PRM method was developed using Skyline (MacCoss Lab, University of Washington). The gel bands were excised, destained, reduced with DTT, alkylated with IAA, and digested with trypsin overnight in 20 mM ammonium bicarbonate. Peptides were extracted three times with 50% acetonitrile in 5% formic acid, then dried by vacuum centrifugation. Samples were then reconstituted in 2% acetonitrile in 0.1% formic acid and injected into an Orbitrap Lumos mass spectrometer using an Easy-nLC chromatography system (Thermo Fisher Scientific). The peptides were eluted over a 90-min gradient and analyzed using the tMS2 method to collect PRM data on the m/z list of precursors. We included four unmodified peptides covering different regions of HTT to normalize the total protein amount. A full scan (MS1) was acquired at 60,000 resolution before targeting the

full list of precursor m/z's. Each targeted precursor m/z was isolated at 0.8 D with an automatic gain control (AGC) target value of 1.5e5 ions and a maximum injection time of 100 ms. The tMS2 fragment ions were acquired at 30,000 resolution. The targeted precursor and fragment ion MS data were aligned by retention time in Skyline and assigned an accuracy score based on the intensities expected from a spectral library created from multiple previously acquired DDA runs. The areas under the curve for the sum of the fragment ions for each precursor m/z were calculated by Skyline and normalized to the unmodified peptides in each sample. Values were reported as fold change for each modified peptide based on the calculated ratio.

### Unbiased screen for HTT interactors upon induction of DSB/DDR

Normal (33CAG) and HD (180CAG) ISPNs were differentiated into medium spiny neuron–like cultures, and DSBs were induced with bleomycin (10 μg/ml, 30 min). HTT protein complexes were pulled down using a mixture of anti-HTT antibodies (MCA2050 and MCA2051) as described above except elution from the beads was performed using 1% SDS/PBS buffer. The relative abundance of coprecipitating with HTT proteins from three control (33CAG) and three HD (180CAG) cell lysates was compared using isobaric mass tags (tandem mass tags, TMT 6-plex, H2A.X; Thermo Fisher Scientific) with nanoflow reversed-phase liquid chromatography–tandem mass spectrometry (RP-nLC-MS/MS) as previously described (35).

Briefly, samples were reduced with dithiothreitol and alkylated with iodoacetamide, then digested with trypsin overnight at 37°C, acidified with TFA, and desalted on a Waters Oasis HLB 96-well plate (30 mg) before isobaric mass labeling (TMT). TMT labeling was performed according to manufacturer's recommendations. TMT-labeled peptides were fractionated on a 1 mm × 10 cm XBridge C18 column.

Peptides were analyzed on an Orbitrap Fusion Lumos mass spectrometer (Thermo Fisher Scientific) interfaced with an Easy-nLC 1200 UPLC by reversed-phase chromatography. Survey scans (MS) of precursor ions were acquired from 350–1,400 m/z at 120,000 resolution. Precursor ions were isolated with a 0.7 m/z window by data-dependent monitoring over 3 s and a 15-s dynamic exclusion. Peptides were fragmented using an HCD activation collision energy 38, an AGC of 1e5, and a maximum injection time of 100 ms at 50,000 resolution.

Fragmentation spectra were processed by Proteome Discoverer v2.4 (PD2.4; Thermo Fisher Scientific) and searched with Mascot v.2.6.2 (Matrix Science) against RefSeq v.83 Human database. Search criteria were as follows: trypsin as the enzyme, one allowed missed cleavage, 5 ppm precursor mass tolerance, 0.02 Da fragment mass tolerance. Modifications included the following: TMT 6-plex on N terminus, carbamidomethylation on C as static, TMT 6-plex on K, oxidation on M, deamidation on N or Q. Peptide identifications from the Mascot searches were processed within PD2.4 using Percolator at a 5% false discovery rate confidence threshold, based on an autoconcatenated decoy database search. Peptide spectral matches (PSMs) were filtered for isolation interference <30%. Relative protein abundance of identified proteins was determined in PD2.4 from the normalized median ratio of TMT reporter ions. The data were normalized by HTT abundance by

Proteome Discoverer (PD2.4) to compensate for minor differences between individual HTT pulldowns. Technical variation in ratios from our mass spectrometry analysis is less than 10% (107).

### APEX2-mediated proximity labeling for HTT interactions

FL-HTT-23Q and polyQ-expanded, FL-HTT-82Q plasmids encoding untagged full-length HTT were described previously (57). APEX2-HTT N-terminal fusions were generated by GenScript. HEK293 cells were transfected with APEX2-HTT-23Q and APEX2-HTT-82Q plasmids (using Lipofectamine 2000; Invitrogen), and APEX2-mediated labeling was performed 24 h after transfection as previously described (77, 78). Briefly, 500 μM biotinyl tyramide (biotin-phenol) (#6241; Tocris) in DMEM (Thermo Fisher Scientific) supplemented with 10% FBS and 1% penicillin–streptomycin was added to all experimental plates except for negative control plates. Labeling was initiated after 1 h by adding H2O2 (1 mM final concentration) for 1 min to all plates. The labeling reaction was quenched by aspirating media from the plate and immediately rinsing three times with the quenching solution: 5 mM Trolox ((+/−)-6-hydroxy-2,5,7,8-tetramethylchromane-2-carboxylic acid [#238813; Sigma-Aldrich]), 10 mM sodium L-ascorbate (#A4034; Sigma-Aldrich), and 10 mM sodium azide in PBS supplemented with protease inhibitors (PIC, Protease Inhibitor Cocktail III, Calbiochem). Cells were then incubated on ice in fresh quenching solution three times for 5 min each. After the last wash, cells were lysed in RIPA buffer (50 mM Tris, pH 7, 150 mM NaCl, 0.1% SDS, 0.5% sodium deoxycholate, 1% Triton X-100) supplemented with protease inhibitors, 10 mM sodium azide, 10 mM sodium ascorbate, and 5 mM Trolox.

200 μl (~200 μg of protein) of cell lysate per sample was reduced, alkylated, digested with trypsin, and desalted as described above, and subsequently dried in a speed vacuum. Dried peptides were solubilized in RIPA buffer and incubated with 300 μl Pierce magnetic streptavidin beads (Thermo Fisher Scientific) overnight at 4° to enrich for biotinylated peptides. After incubation, the beads were washed sequentially with the following buffers to remove nonspecific proteins: twice with RIPA buffer, once with 1 M KCl, once with 0.1 M Na₂CO₃, twice with RIPA buffer, and twice with PBS. After the last wash, biotinylated peptides were eluted from streptavidin beads by incubation with 600 μl of neat hexafluoroisopropanol (HFIP) for 5 min at RT. After second elution, the supernatants were combined into one tube and then dried in a speed vacuum. The dried peptides were processed for mass spectrometry after TMT labeling as described above.

Fragmentation spectra were processed by Proteome Discoverer v2.4 (PD2.4; Thermo Fisher Scientific) and searched as above with Mascot v.2.6.2 (Matrix Science) against RefSeq v.83 Human database including custom APEX2 sequence (L-ascorbate peroxidase APx2 Q39843_SOYBN). Modifications included are the same as above, plus biotinyl-tyramide. The data were normalized by the abundance of the APEX2 protein by Proteome Discoverer (PD2.4).

### Graphs and pathway and process enrichment analysis

PCA plots, heat map, and volcano plots were built by PD2.4 using normalized abundance of proteins. Pathway and process enrichment

analyses were performed using "Metascape" (65). Protein–protein interaction (PPI) enrichment analyses were generated by "Metascape" using following databases: STRING6, BioGrid7, OmniPath8, InWeb_IM9. Only physical interactions in STRING (physical score > 0.132) and BioGrid were used. The resultant networks contain the subset of proteins that form physical interactions with at least one other member in the list. If the network contains between three and 500 proteins, the Molecular Complex Detection (MCODE) algorithm 10 has been applied to identify densely connected network components. Pathway and process enrichment analyses have been applied to each MCODE component independently, and the best-scoring terms by *P*-value have been retained as the functional description of the corresponding components, shown in the tables underneath corresponding network plots within Figs 4 and 10.

## Immunofluorescence (IF) and proximity ligation assay (PLA)

Immunofluorescence (IF) was carried out using standard methods after the fixation of the cells with 4% PFA. Cells were permeabilized with 0.1% Triton X-100 for 10 min, blocked with 5% goat serum, and incubated with primary antibodies ON at $4^0$C. The following primary antibodies were used for staining: phospho-histone H2A.X (S139) (3F2, mouse) from Genetex (#GTX80694) at 1:200; phospho-histone H2A.X (S139) (20E3, rabbit mAb) from Cell Signaling Technology (#60566) at 1:200; phospho-DNA-PKcs (S2056) from Abcam (#18192) at 1:400; DNA-PKcs (3H6, mouse) from Cell Signaling Technology (#12311) at 1:100; SC35 (SC-35) from Sigma-Aldrich (#S4045) at 1:1,000; TCERG1/CA150 from Invitrogen (#PA5-84824) at 1:300; BRG1 (A52) from Cell Signaling Technology (#3508) at 1:100; ARID1A (EPR13501) from Abcam (#182560) at 1:500; and MCA2050 (at 1:100) and S1181 (at 1:300, described above) for HTT. Secondary antibodies used in this study were as follows: goat anti-rabbit IgG Alexa Fluor 488 and goat anti-mouse IgG Alexa Fluor 555 (Thermo Fisher Scientific). Images were acquired with a Zeiss LSM 700 confocal microscope with NA 0.55 condenser using 63x/1.4 PlanApo oil, with DIC objective (Carl Zeiss Microscopy, LLC).

We performed quantitation of γ-H2A.X foci, and of nuclear puncta positive for pS1181-HTT or for SC35 using Imaris (Bitplane AG) and Huygens Essential (Scientific Volume Imaging B.V., www.svi.nl) software. The confocal parameters were determined, and the same settings were applied for each sample within the dataset. The images with z-stacks (step size ~0.3 μm) were collected using Zen image collection software (Carl Zeiss Microscopy) and processed for 3D reconstruction. For correct measurement of colocalization, preliminary image deconvolution was performed to eliminate background and noise of the fluorescent signal. We have chosen nucleus as a region of interest for individual cells and made a 3D surface of nucleus using the "DAPI" channel. Analysis was performed on five images per group (taken from independent wells) with 7-12 cells per image.

PLA was performed using Duolink In Situ Mouse/Rabbit Kit (MilliporeSigma) according to the manufacturer's protocol and using primary antibodies described above. PLA signals were quantitated using MetaXpress software (Molecular Devices). The number of PLA sites per cell, sum intensity of PLA sites per cell, and average PLA site intensity were reported relative to technical negative single antibody control within a given experiment. Three to four images were taken from independent wells; a total of 30–50 cells were analyzed for each condition.

## Validation of the antibodies used in the study

All antibodies to phospho-HTT have been generated in Seong laboratory, and their specificity has been previously validated by dot blot with phosphorylated and nonphosphorylated peptides as a negative control (58). In addition, we have also previously validated these antibodies in our laboratory using phosphorylation-null HTT constructs (Fig S2).

To validate TCERG1, BRG1, and ARID1A antibodies, HEK293 cells were transfected with plasmids encoding tagged TCERG1 (T7-tag), BRG1 (FLAG tag), or ARID1A (V5-tag). Western blotting was performed with total cell lysates using antibodies to tags (Fig S6A). We observed protein-specific bands in the lysates of transfected cells, but not in nontransfected control samples. Probing of parallel blots with corresponding protein-specific antibodies produced protein bands of similar size with a substantial increase of detected protein levels in transfected cells compared with endogenous proteins, which were also detected in some samples. The following plasmids were used: pEF-BOS-T7-CA150 was a gift from Mariano Garcia-Blanco (plasmid # 21924; Addgene; http://n2t.net/addgene:21924; RRID:Addgene_21924) (108); pCMV5 BRGI-Flag was a gift from Joan Massague (plasmid # 19143; Addgene; http://n2t.net/addgene:19143; RRID: Addgene_19143) (109); and pcDNA6-ARID1A was a gift from Ie-Ming Shih (plasmid # 39311; Addgene; http://n2t.net/addgene:39311; RRID: Addgene_39311) (110).

We have validated anti-phospho DNA-PKcs (S2056) antibody (#18192; Abcam) and total DNA-PKcs (E6U3A) antibody (#38168; Cell Signaling Technology) in our ISPNs using DNA-PKcs knockdown to confirm specificity (Fig S6B and C): ISPNs were transfected at a final concentration of 1.5uM with siRNA to DNA-PKcs (Santa Cruz Biotechnology) or with scrambled control siRNA. Cells were electroporated with one 30-ms pulse at 620V using the SP100 electroporator (Celetrix 11-0103). Cells were then cultured for 24 h in SCM-2 media supplemented with tamoxifen. After 24 h, cells were grown in SCM-2 without tamoxifen for 48 h. Using Western blot with antibodies either to total or phospho DNA-PKcs, we found that the efficiency of the knockdown was 40–80% (Fig S6B). To validate phospho-DNA-PKcs antibody in IF, we performed staining of ISPNs after transfection with either DNA-PKcs or scrambled control siRNA (Fig S6C). As shown, levels of pDNA-PKcs, detected with this antibody, were substantially lower after the knockdown, whereas levels of SC35 (used here as a reference protein) remained unaffected. In addition, anti-phospho DNA-PKcs (S2056) antibody (#18192; Abcam) was validated by the manufacturer using knockdown to confirm specificity (https://www.abcam.com/en-us/products/primary-antibodies/dna-pkcs-phospho-s2056-antibody-ab18192). Total DNA-PKcs (E6U3A) antibody (#38168; Cell Signaling Technology) was validated by the manufacturer using knockdown to confirm specificity in both Western blotting and IF (https://www.cellsignal.com/products/primary-antibodies/dna-pkcs-e6u3a-rabbit-mab/38168).

## Statistical analysis

Statistical analysis was performed using SigmaPlot software. The data were first checked using the Shapiro–Wilk normality test and equal variance test. If passed, a *t* test with equal variances was performed for two group comparisons. If normality or equal variance test failed, the Mann–Whitney rank sum test was performed. For multiple group comparisons, we used one-way analysis of variance (ANOVA) with pairwise multiple comparison procedures (Holm–Sidak's method). Graphs showing individual experimental values were built using Prism (GraphPad). In the experiments involving cell imaging, each independent image/plate/well was assumed as a biological replicate. In the experiments involving evaluation of protein expression, each individual experiment was assumed as a biological replicate. The statistical tests used in each figure, number of biological replicates for each assay, and all *P*-values for statistically significant results are noted in the figure legends.

# Data Availability

All data are available from the corresponding authors upon reasonable request. The mass spectrometry proteomics data have been deposited to the ProteomeXchange Consortium (http://proteomecentral.proteomexchange.org) via the PRIDE partner repository with the dataset identifiers: PXD059660 (IP-MS study) and PXD059663 (APEX2-proximity study).

# Supplementary Information

# Acknowledgements

This work was supported by National Institutes of Health: 5R21NS109412-02 to T Ratovitski and 5R01NS08645208 to CA Ross, and by the JHU HD Precision Medicine Center of Excellence. We thank Tom Vogt and Tapas Hazra for discussions.

## Author Contributions

T Ratovitski: conceptualization, data curation, formal analysis, supervision, funding acquisition, investigation, methodology, project administration, and writing—original draft.
CD Holland: validation, investigation, and methodology.
RN O'Meally: formal analysis, validation, investigation, and methodology.
AV Shevelkin: formal analysis, validation, investigation, and methodology.
SV Kamath: investigation and methodology.
T Shi: investigation and methodology.
MJ Rodriguez: validation, investigation, and methodology.
RN Cole: data curation, formal analysis, supervision, methodology, project administration, and writing—review and editing.
M Jiang: supervision, investigation, methodology, and writing—review and editing.
CA Ross: conceptualization, resources, supervision, funding acquisition, project administration, and writing—review and editing.

## Conflict of Interest Statement

The authors declare that they have no conflict of interest.

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
