## [Reviewer comments · Life Science Alliance]

Huntingtin (HTT) interactome in regulation of DNA repair/remodeling and RNA processing pathways

Tamara Ratovitski, Chloe Holland, Robert O'Meally, Alexey Shevelkin, Siddhi Kamath, Tianze Shi, Matthew Rodriguez, Robert Cole, Mali Jiang, and Christopher Ross

DOI: <https://doi.org/10.26508/lsa.202503424>

Corresponding author(s): Tamara Ratovitski, Johns Hopkins University and Christopher Ross, Johns Hopkins

Review Timeline:

Submission Date:	2025-06-18
Editorial Decision:	2025-09-08
Revision Received:	2025-11-20
Editorial Decision:	2025-12-18
Revision Received:	2026-02-01
Editorial Decision:	2026-02-24
Revision Received:	2026-03-02
Accepted:	2026-03-04

Scientific Editor: Tim Fessenden

Transaction Report:

September 8, 2025

Re: Life Science Alliance manuscript #LSA-2025-03424

Dr. Tamara Ratovitski
Johns Hopkins University
600 North Wolfe Street
CMSC 5 South
Baltimore 21287

Dear Dr. Ratovitski,

Thank you for submitting your manuscript entitled "Huntingtin (HTT) interactome in regulation of DNA repair/remodeling and RNA processing pathways" to Life Science Alliance. The manuscript was assessed by expert reviewers, whose comments are appended to this letter. We appreciate your patience during the unusually long duration of the peer review process.

As you will see, reviewers agreed that the proteomic profiling of HTT in the context of DNA repair marks a valuable contribution, although they differed in their overall level of enthusiasm. We concur with Reviewer 1 that claims made on pS1181, TCERG1, and DNA-PKcs, which rely heavily on imaging and PLA, should be validated in a manner of your choosing. We also agree that the discordant results on HTT localization should be more directly confronted, whether with new data or with changes to the text. Confirming the functional relevance of these findings in a manner suggested by Reviewer 1 would also strengthen this work. The quality of the imaging results was a concern of Reviewer 2 and we invite you to consider their several suggestions to improve these. We agree that the contrast settings of images in Figures 6D, 7B, 9B, and 10B should be adjusted for clarity. Additional requests besides those mentioned above are not required in a revision.

Finally, because LSA requires that all claims are supported with data, please remove the claim related to Figure 2B on subnuclear structures or else provide data supporting this.

Thank you for this interesting contribution to Life Science Alliance. We are looking forward to receiving your revised manuscript.

Sincerely,

- A letter addressing the reviewers' comments point by point.
- An editable version of the final text (.DOC or .DOCX) is needed for copyediting (no PDFs).
- High-resolution figure, supplementary figure and video files uploaded as individual files: See our detailed guidelines for

preparing your production-ready images, <https://www.life-science-alliance.org/authors>

B. MANUSCRIPT ORGANIZATION AND FORMATTING:

Reviewer #1 (Comments to the Authors (Required)):

This is an interesting manuscript that addresses the nuclear functions of huntingtin (HTT) and its potential role as a scaffold connecting DNA repair, chromatin remodeling, and RNA processing. The authors make use of HD patient-derived striatal precursor neurons and apply a wide range of techniques including immunofluorescence, co-immunoprecipitation, PLA, quantitative proteomics, and proximity labeling. They report that HD neurons mount a defective γ H2AX response to double-strand breaks, that phosphorylation of HTT at S1181 is induced by DNA damage and mediated in vitro by DNA-PK, and that phosphorylated HTT localizes to nuclear speckles. They further identify interactions with proteins such as TCERG1 and subunits of the Mediator and BAF complexes, and employ APEX2 labeling to show enrichment of RNA-processing and chromatin factors around HTT.

The topic is timely, as there is growing appreciation that HTT exerts nuclear functions relevant to pathogenesis, and the focus on nuclear speckles is particularly intriguing. The proteomics dataset is a strength, with appropriate replication, public deposition, and overlap with known HD modifiers. The nuclear speckle angle is compelling, and targeted perturbations-for example, splicing reporters or minigenes-could directly test whether HTT-DNA-PK signaling feeds into RNA-processing outcomes under genotoxic stress.

Despite these strengths, several issues temper enthusiasm. A major concern is the heavy reliance on antibody-based assays without rigorous controls. Antibodies to phospho-HTT and many interactors are not validated by siRNA/CRISPR knockdown, peptide competition, or similar approaches. Given well-known problems with phospho-epitope specificity, this is a serious weakness. Much of the evidence, especially in the pS1181, TCERG1, and DNA-PKcs figures, depends on immunofluorescence and PLA signals that remain difficult to interpret without such validation. Appropriate remedies would be to validate antibodies directly (e.g., peptide competition, knockdown/CRISPR loss-of-signal), include genetic controls for interactors such as PRKDC, TCERG1, and BRG1/ARID1A, and add pharmacological inhibition to separate DNA-PK from ATM contributions to pS1181 and γ H2AX signaling under ROS, IR, or bleomycin treatment.

A second limitation is that the work is largely observational. The authors demonstrate co-localization and interaction, but the mechanistic consequences are not explored. Mutating S1181, or deploying DNA repair reporters, could establish whether the described interactions affect DNA repair capacity or RNA splicing outcomes. Useful additions would be endogenous S1181A/S1181D knock-ins or CRISPRi/a at this site; assays of NHEJ efficiency (e.g., EJ5-GFP or traffic-light reporters) and survival after DNA damaging agents; and splicing readouts such as minigene assays or RNA-seq, ideally combined with kinase inhibition.

The proteomics results are convincing, but a key discrepancy arises: mass spectrometry suggests increased interaction of HTT with BAF/Mediator subunits in HD neurons, while PLA shows decreased nuclear proximity. The authors attribute this to nuclear depletion of HTT. This explanation could be tested in several ways, and any one would suffice. The most direct approach would be subcellular fractionation followed by HTT IP/MS (or immunoblot) using validated nuclear and cytoplasmic markers, to determine whether interactions are reduced in nuclear fractions but increased in cytoplasmic/perinuclear fractions of HD neurons. Alternative strategies include CRISPR-based endogenous tagging of HTT for live or super-resolution imaging, compartment-restricted proximity assays such as nuclear PLA, CUT&Tag, or BioID, or a nuclear rescue experiment with NLS-

HTT. Each would clarify whether loss of nuclear HTT underlies the reduced PLA despite higher bulk recovery.

Finally, the APEX2 labeling provides useful information, but the minimal differences between wild-type and expanded HTT highlight the need for endogenous tagging strategies to reduce overexpression artefacts. For the pS1181 and DNA-PK figures, the inclusion of selective inhibitors would be especially helpful. DNA-PKcs inhibitors such as AZD7648 (0.5-2 μ M) or ATM inhibitors such as AZD0156 (50-200 nM) given prior to damage would test causality, and when combined with PRKDC or ATM knockdown would validate kinase assignment and reconcile MS vs PLA discrepancies by revealing compartment-specific regulation.

The study brings together a substantial dataset and adds to our understanding of HTT's nuclear interactome, but I am less enthusiastic about the reliability of the antibody-based data and the lack of mechanistic follow-through. I would recommend major revisions focused on three aspects: (1) proper antibody validation controls, (2) inclusion of at least one functional assay directly linking HTT modifications or interactions to DDR or RNA processing, and (3) clarification of discrepancies between different assay readouts. With these additions the paper would make a strong contribution to the field; as it stands, it is descriptive and somewhat vulnerable to technical artefacts.

Reviewer #2 (Comments to the Authors (Required)):

This manuscript is defining the interactions of huntingtin in DNA damage repair. They have identified the direct signaling on huntingtin at S1181 by DNA-PK kinase. By cell biology, they see huntingtin at nuclear puncta with TCERG1 and MED15, both defined as modifiers of HD by GWAS. The strength of this manuscript is the mass spectrometry interactome data, which has not been done in this detail in HD models in the context of DDR.

Major points: The confocal imaging data throughout the manuscript is problematic. Some of the westerns are over saturated. The manuscript could benefit from a re write with fewer figures and less reliance on co-localization by fluorescence microscopy. They justify the use of an extremely long CAG180 allele IPSNs (which is likely not stable) and compare to a wild type allele of CAG 33. Yet, no mutant length allele prior to this >150 CAG threshold was used, which is a major omission. This implies that DSBs are not being properly repaired at >CAG150 but we have no data from pre-expansion mutant alleles.

Minor points:

Abstract should remove "(in vitro)", it's a detail not relevant to an abstract. Reference to the McCarroll lab in intro should mention that the role of somatic expansion in HD is not known, there is no mechanism, and currently this is just correlation. Recent publication from HD GEM consortium revealed that the most potent gene modifier of HD, CAA allele interruptions, does not influence somatic expansion. While there is not idea why CAA alleles modify HD, it's not at the level of CAG stability.

CAG180 is a very long expansion model. There should have been a early HD allele used around CAG43, the mean clinical allele length.

"Thus, gH2A.X is a sensitive molecular marker of both DNA damage and repair (48)." This is not exactly accurate, it is a marker of DDR response, but not a measure of DNA damage.

Figure 1 could have been prepared with more care. 1C is showing the reader excluded data, so not sure what the point is in making it a figure. Also, image boxes are not aligned and text is the wrong case. 1A is qualitative, and the thresholding of signal is arbitrary. If they want to show more DSBs, they should image with P53BP1. Unclear how they actually used Imaris software. There is no mention if CAG 33 and CAG180 IPSNs are isogenic or not.

Figure 2B and throughout the figures should have color changed to color safe standards of green, magenta, white. There should be a zoomed image to make the point that the gHRaX puncta are not HTT S-1181-P04 puncta.

Figure 3A: were these confocal images rasterized sequentially or with both 488 and 543 lasers on at the same time? No need for the three channel merged panels, SC35 is only nuclear.

Figure 4 A images are over saturated, imaging should be done at a lower intensity levels and higher magnification, and should benefit from some type of super-resolution imaging. You get a lot of yellow with a lot of red and green, to make sure this isn't artifactual they should do a Pearson's correlation and Mander's overlap analysis. The PLA data in 4C would be clearer without the blue channel.

Figure 5E. Not much point to the graphics when the labels are so crowded they are illegible.

Figure 6D, again, channels are over saturated. I would suggest getting advice from a technician of the imaging facility in use here.

Figure 6C, while we see clearly elevated S1181-PO4 with time, We cannot conclude anything from the other westerns because they are clearly saturated at time zero.

Figure 7B. Same problem with over-saturation of pixels as in Figure 6. This comes up again in Figure 9 and 10 and all supplemental microscopy images.

9B: see comment of figure 4.

Given co-localization by microscopy tells us nothing about direct interaction, but only regional vicinity, the manuscript could be improved by removing all the over saturated confocal images and just presenting the PLA data.

Response to Reviewer #1

Comment

The topic is timely, as there is growing appreciation that HTT exerts nuclear functions relevant to pathogenesis, and the focus on nuclear speckles is particularly intriguing. The proteomics dataset is a strength, with appropriate replication, public deposition, and overlap with known HD modifiers. The nuclear speckle angle is compelling, and targeted perturbations—for example, splicing reporters or minigenes—could directly test whether HTT-DNA-PK signaling feeds into RNA-processing outcomes under genotoxic stress.

Response

We thank the reviewer for asserting the significance of our study and for raising important questions and helpful suggestions for the future directions.

Comment

A major concern is the heavy reliance on antibody-based assays without rigorous controls. Antibodies to phospho-HTT and many interactors are not validated by siRNA/CRISPR knockdown, peptide competition, or similar approaches. Given well-known problems with phospho-epitope specificity, this is a serious weakness. ...immunofluorescence and PLA signals that remain difficult to interpret without such validation. Appropriate remedies would be to validate antibodies directly (e.g., peptide competition, knockdown/CRISPR loss-of-signal), include genetic controls for interactors such as PRKDC, TCERG1, and BRG1/ARID1A, and add pharmacological inhibition to separate DNA-PK from ATM contributions to pS1181 and γ H2AX signaling under ROS, IR, or bleomycin treatment.

Response

This point was well taken. We agree that both co-IP and PLA methods rely on the antibodies (although in different ways), but these are the most widely used techniques to assess protein interactions. We agree that only validated antibodies should be used, especially when it comes to phospho-specific antibodies. For clarity, we have now added a more detailed description of the antibody validation with corresponding references in the text and **the new section of the Methods (page 23)**. Below is some additional information for the reviewer on the validation of antibodies used in this study:

- As described in **the new section of the Methods (page 23)**, all antibodies to phospho-HTT (generated in Seong lab, Jung et al, 2020) has been previously validated by dot blot with phosphorylated and non-phospho peptides as negative control:

[Figure removed by editorial staff per authors' request]

-In addition, we have also previously validated these antibodies using phosphorylation null HTT constructs (Ratovitski et al, 2017):

[Figure removed by editorial staff per authors' request]

-We have now also included **new data** for validation of TCERG1, BRG1, and ARID1A antibodies using tagged expression plasmids (**new Fig. S3**)

-Notably, anti-phospho DNA- PKcs (S2056) antibody (Abcam #18192) was validated by the manufacturer using knock down to confirm specificity <https://www.abcam.com/en-us/products/primary-antibodies/dna-pkcs-phospho-s2056-antibody-ab18192>.

[Figure removed by editorial staff per authors' request]

-Total-DNA-PKcs [E6U3A] antibody (Cell Signaling Technology #38168) was validated by the manufacturer using knock down to confirm specificity

Comment

A second limitation is that the work is largely observational. The authors demonstrate co-localization and interaction, but the mechanistic consequences are not explored. Mutating S1181, or deploying DNA repair reporters, could establish whether the described interactions affect DNA repair capacity or RNA splicing outcomes. Useful additions would be endogenous S1181A/S1181D knock-ins or CRISPRi/a at this site; assays of NHEJ efficiency (e.g., EJ5-GFP or traffic-light reporters) and survival after DNA damaging agents; and splicing readouts such as minigene assays or RNA-seq, ideally combined with kinase inhibition.

Response

We thank the reviewer for excellent suggestions for future studies, such as S1181A/S1181D knock-ins or CRISPR followed by functional assays for DNA repair, etc. These studies are under consideration but are not feasible within the time frame of this revision.

Comment

The proteomics results are convincing, but a key discrepancy arises: mass spectrometry suggests increased interaction of HTT with BAF/Mediator subunits in HD neurons, while PLA shows decreased nuclear proximity.

The authors attribute this to nuclear depletion of HTT. The most direct approach would be subcellular fractionation followed by HTT IP/MS (or immunoblot) using validated nuclear and cytoplasmic markers, to determine whether interactions are reduced in nuclear fractions but increased in cytoplasmic/perinuclear fractions of HD

Response

The **discrepancy between our co-IP and proximity -based analyses** of HTT/BAF interactions have been discussed in detail in the **expanded section of Discussion, page 15)** in light of different aspects of methodologies of these two approaches.

We have followed up on the reviewer's suggestion and attempted subcellular fractionation, but IP from nuclear fractions turned out to be not feasible due to very low recovery of nuclear HTT after biochemical fractionation.

We have taken an alternative approach- to quantify the changes in cytoplasmic/nuclear distribution of PLA signals. Our ARID1A/HTT proximity analysis showed a dramatic boost in cytoplasmic/nuclear ratios for PLA sum intensity and average PLA site intensity in HD ISPNs compared to control neurons (**new Fig. 10E**), This shift towards cytoplasmic proximity in combination with depleted nuclear HTT may in part account for decreased nuclear PLA signals in HD neurons, at least for ARID1A/HTT proximity. We want to remind the reviewer that in situ proximity-based method (PLA) can preserve the intracellular spatial information of the interactomes and its amplified readout signal is highly dependent on the levels of the protein of interest in specific subcellular locations and consequently the number of antibody's molecules bound to the target.

Comment

Alternative strategies include CRISPR-based endogenous tagging of HTT for live or super-resolution imaging, compartment-restricted proximity assays such as nuclear PLA, CUT&Tag, or BioID, or a nuclear rescue experiment with NLS-HTT. Each would clarify whether loss of nuclear HTT underlies the reduced PLA despite higher bulk recovery.

Response

We agree with the reviewer that CRISPR-based endogenous tagging of HTT for live imaging has a great potential for follow-up studies of HTT subcellular localization and interactions. These studies are under way but are independent and beyond the scope of the current revision

Comment

Finally, the APEX2 labeling provides useful information, but the minimal differences between wild-type and expanded HTT highlight the need for endogenous tagging strategies to reduce overexpression artefacts.

Response

We completely agree as noted in the current version: “To overcome this difficulty, it would be beneficial to engineer APEX2 tag on the endogenous HTT (e.g. by CRISPR) in the future studies. APEX2 tag can also be placed at the C-terminus or within other portions of HTT to further explore domain-specific interactions”.

Comment

For the pS1181 and DNA-PK figures, the inclusion of selective inhibitors would be especially helpful. DNA-PKcs inhibitors such as AZD7648 (0.5-2 μ M) or ATM inhibitors such as AZD0156 (50-200 nM) given prior to damage would test causality, and when combined with PRKDC or ATM knockdown would validate kinase assignment and reconcile MS vs PLA discrepancies by revealing compartment-specific regulation.

Response

We thank the reviewer again for these great suggestions. We are currently pursuing DNA-PK chemical inhibition and DNA-PKcs knockdown studies in a parallel independent project

Response to Reviewer #2

Comment

This manuscript is defining the interactions of huntingtin in DNA damage repair. They have identified the direct signaling on huntingtin at S1181 by DNA-PK kinase. By cell biology, they see huntingtin at nuclear puncta with TCERG1 and MED15, both defined as modifiers of HD by GWAS. The strength of this manuscript is the mass spectrometry interactome data, which has not been done in this detail in HD models in the context of DDR.

Response

We thank the reviewer for careful consideration and for noticing a few overlooked issues that now have been corrected

Comment

The confocal imaging data throughout the manuscript is problematic. Some of the westerns are over saturated. The manuscript could benefit from a re write with fewer figures and less reliance on co-localization by fluorescence microscopy.

Response

We have adjusted all co-localization images to lower intensity (to remove saturated pixels) and higher magnification and adjusted over saturated westerns. We have also included new quantitative data of co-localization with Pearson correlation and Manders overlap analyses (**new panels 3B, 4B, 7C, 9B, 10B**)

Comment

They justify the use of an extremely long CAG180 allele IPSNs (which is likely not stable) and compare to a wild type allele of CAG 33. Yet, no mutant length allele prior to this >150 CAG threshold was used, which is a major omission. This implies that DSBs are not being properly repaired at >CAG150 but we have no data from pre-expansion mutant alleles.

Response

As recently shown by McCarroll lab (and as noted in the Introduction), MSNs with up to 150 CAG repeats have few changes in gene expression, while neurons with greater than 150 CAGs have loss of important MSN-appropriate, cell-type-defining messages and inappropriate expression of messages related to development or cell toxicity. This justifies our use of CAG180 model to study altered interactions of mHTT protein as potential consequence of altered gene expression most relevant to HD pathogenesis.

However we agree with the reviewer that analysis of intermediate CAG models could be extremely useful, and we have recently developed ISPNs with CAG 43, 60 and 77 which are currently being characterized.

Comment

CAG180 is a very long expansion model. There should have been an early HD allele used around CAG43, the mean clinical allele length

Response

The reviewer is right that HD CAG repeat lengths associated with adult onset measured in blood DNA is around 39-43 CAG for full penetrance of HD. Since inherited tract expands through life and most subjects develop disease in mid-life, this implies that in degenerating neurons the CAG length is likely to be longer than the inherited length. In fact, as shown by Heintz and McCarroll labs, due to somatic expansion MSNs have far more expansions to very long repeat lengths (150+ CAG) than other neuron types, and this is central to HD pathogenesis.

In light of these recent findings and current hypothesis in the field, our immortalized striatal precursor neurons (ISPN) differentiated to a phenotype resembling Medium Spiny Neurons with a repeat of 180CAG is likely very relevant to HD pathogenesis

Minor points:

Comment

Abstract should remove "(in vitro)" , it's a detail not relevant to an abstract.

Response

Removed

Comment

Reference to the McCarroll lab in intro should mention that the role of somatic expansion in HD is not known, there is no mechanism, and currently this is just correlation.

Recent publication from HD GEM consortium revealed that the most potent gene modifier of HD, CAA allele interruptions, does not influence somatic expansion. While there is not idea why CAA alleles modify HD, it's not at the level of CAG stability.

Response

We thank the reviewer for this helpful comment. The following paragraph was included in the Introduction:

“Although recent studies suggest that somatic expansion may be an important driver of pathology (44, 45), the mechanism is not known. The most recent GWAS study identified several HD modifiers not obviously involving DNA repair (MED15, RRM2B, CCDC82 and TCERG1) (47). The same study found that non-canonical HTT CAG repeat sequences (CAA.CCA) modify motor onset but does not increase HTT CAG repeat expansion, suggesting a different mechanism from the initial somatic expansion phase”.

Comment

"Thus, gH2A.X is a sensitive molecular marker of both DNA damage and repair (48)." This is not exactly accurate, it is a marker of DDR response, but not a measure of DNA damage.

Response

Corrected

Comment

Figure 1 could have been prepared with more care. 1C is showing the reader excluded data, so not sure what the point is in making it a figure. Also, image boxes are not aligned and text is the wrong case. 1A is qualitative, and the thresholding of signal is arbitrary. If they want to show more DSBs, they should image with P53BP1. Unclear how they actually used Imaris software.

Response

We have realigned image boxes.

The use of Imaris software is described in detail in a paragraph in the Methods section (p 23, line)

Comment

There is no mention if CAG 33 and CAG180 ISPNs are isogenic or not.

Response

No. As described in Methods, Immortalized Striatal Precursor Neurons (ISPNs) were generated in our laboratory from patient-derived iPS cells with normal (33) or expanded (180) CAG repeats. The reference is given

Comment

Figure 2B and throughout the figures should have color changed to color safe standards of green, magenta, white. There should be a zoomed image to make the point that the gHRaX puncta are not HTT S-1181-P04 puncta.

Response

We have included zoomed images on revised Fig. 2B

Comment

Figure 3A: were these confocal images rasterized sequentially or with both 488 and 543 lasers on at the same time?

Response

Yes, as noted in Methods, Images were acquired with Zeiss LSM 710 confocal microscopes with sequential rasterization

Comment

Figure 4 A images are over saturated, imaging should be done at a lower intensity levels and higher magnification, and should benefit from some type of super-resolution imaging . You get a lot of yellow with a lot of red and green, to make sure this isn't artifactual they should do a Pearson's correlation and Mander's overlap analysis. The PLA data in 4C would be clearer without the blue channel.

Response

We have adjusted co-localization images on Fig. 4A to lower intensity (to remove saturated pixels) and higher magnification. We thank the reviewer for a good suggestion to include quantitation with Pearson and Manders co-localization coefficients. It is now included **in new panel 4B**

Comment

Figure 6D, again, channels are over saturated. I would suggest getting advice from a technician of the imaging facility in use here.

Response

Fig. 6D was removed

Comment

Figure 6C, while we see clearly elevated S1181-PO4 with time, We cannot conclude anything from the other westerns because they are clearly saturated at time zero.

Response

We have adjusted images on Fig. 6C (now Fig, 6B)

Comment

Figure 7B. Same problem with over-saturation of pixels as in Figure 6. This comes up again in Figure 9 and 10 and all supplemental microscopy images.

Response

We have adjusted all co-localization images to lower intensity (to remove saturated pixels) and higher magnification. We have also included new quantitative data of co-localization with Pearson correlation and Manders overlap analyses (**new panels 7C, 9B, 10B**)

Comment

Given co-localization by microscopy tells us nothing about direct interaction, but only regional vicinity, the manuscript could be improved by removing all the over saturated confocal images and just presenting the PLA data.

Response

We thank the reviewer for his suggestion to conduct Pearson's correlation and Mander's overlap analysis for co-localization. We feel that with this addition co-localization data is well validated and provide useful information about reduced co-localization in HD cells in some cases.

December 18, 2025

Re: Life Science Alliance manuscript #LSA-2025-03424R

Dr. Tamara Ratovitski
Johns Hopkins University
600 North Wolfe Street
CMSC 5 South
Baltimore 21287

Dear Dr. Ratovitski,

Thank you for submitting your revised manuscript entitled "Huntingtin (HTT) interactome in regulation of DNA repair/remodeling and RNA processing pathways" to Life Science Alliance. We sought to return the revised work to Reviewer 1 but unfortunately they were unavailable. Because resolution of their concerns was required for further consideration at LSA, we sought expert input from a fourth reviewer. This new reviewer appreciated the potential importance of these findings, while also confirming that the requests from Reviewer 1 on controls and validation assays were indeed essential to support the main claims in this study. As you will see, Reviewer 4 notes that these issues remain unresolved in this revision. They stated to the editors that while the work is interesting, the issues with lack of controls and questions on rigor preclude them from recommending publication.

Our policy is that papers are considered through only one revision cycle. However given our interest in this work, the support from Reviewer 3, and the resolution of concerns raised by Reviewer 2, we are open to making an exception and granting one additional short round of revision. If you wish to pursue consideration of this work, please furnish a revised manuscript that includes the validation assays requested by Reviewers 1 and 4 on phospho-DNA-PKcs and appropriate negative controls for the PLA assays. Related to concerns about the PLA data, we also concur with this reviewer that the significance of the data in Figures 6D and 10E is unclear. Please approach these required validation assays in the manner described by Reviewer 4. This reviewer also raised concerns on the proteomics data in the final figure consistent with prior concerns by Reviewer 1. We feel the main advance in this work is supported without this final dataset and we concur it should be removed or, if you wish, retained and explicitly framed as preliminary. Finally, the request from Reviewer 1 and reiterated by Reviewer 4 on NHEJ was another point that we suggested should be addressed if possible. In the absence of these data, the text must be revised (in the abstract, results, and discussion) to acknowledge that the physiological outcome of HTT interaction with DNA-PKcs shown here remains to be examined.

If you wish to pursue publication at LSA, please submit the final revision along with a letter that includes a point by point response to the remaining reviewer comments.

To upload the revised version of your manuscript, please log in to your account: <https://lsa.msubmit.net/cgi-bin/main.plex>
You will be guided to complete the submission of your revised manuscript and to fill in all necessary information.

B. MANUSCRIPT ORGANIZATION AND FORMATTING:

Sincerely,

Reviewer #4 (Comments to the Authors (Required)):

The revised manuscript by Ratovitski et al. has addressed some of the concerns from previous reviewer's while several still remain. Importantly, the question of antibody specificity has not been adequately addressed. While manufacturers may have validate these antibodies to a certain extent, many of them were done by western blotting while the authors here use these antibodies in immunofluorescence. This is a completely different technique and fixation methods. Western blotting uses denatured proteins while IF fixes proteins and the signal of an antibody can be very different. This still remains a concern. For example, in Figure 7C, the pDNA-PKcs antibody is used. This should be shown to be dependent on DNA-PKcs total protein, it also should be DNA-PK dependent and an IF with unmodified DNA-PKcs showing similar localization would be beneficial. It is well known in the field that phosphorylation antibodies can detect other sites on proteins and in IF, their isn't the size information that you get by Western blotting to rule out signals on other differently sized proteins. Fig 6D is also problematic since the PLA signal gives so my cytoplasmic signal, where you wouldn't expect these proteins to be. Standard controls for PLA experiments is the use of single antibody reactions so that these spurious signals can be understood and ruled out. Figure 10e is misleading and should be removed. Fig 11 adds little to nothing to the manuscript as presented. It is unclear where the data is coming from for the 4 conditions tested and it appears that this experiment may not have even worked for technical issues. It is unclear what additional value this experiment adds. If the authors can reexam the data and present it in a manner that is more informative for the reader, then this could be considered but as is, Fig 11 is uninformative and should be removed. Finally, a reduction in DNA damage signaling does not address the question of whether or not this is biologically significant. It was asked to look directly at NHEJ in these cells that display reduced DDR and this is still a valid experiment that should be performed prior to publication. If not, the significance of altered DDR and interaction with DNA-PKcs by HTT in 180CAG cells remains uncertain.

Response to Reviewer #4

We thank the reviewer for reevaluating the manuscript in light of previous comments and revision.

Comment

...the question of antibody specificity has not been adequately addressed. While manufacturers may have validate these antibodies to a certain extent, many of them were done by western blotting while the authors here use these antibodies in immunofluorescence. This is a completely different technique and fixation methods. Western blotting uses denatured proteins while IF fixes proteins and the signal of an antibody can be very different. This still remains a concern. For example, in Figure 7C, the pDNA-PKcs antibody is used. This should be shown to be dependent on DNA-PKcs total protein, it also should be DNA-PK dependent

For example, in Figure 7C, the pDNA-PKcs antibody is used. This should be shown to be dependent on DNA-PKcs total protein, it also should be DNA-PK dependent

Response

We agree with the reviewer that antibody binding to the target maybe different in denaturing (SDS western blot) conditions vs immunofluorescence. Of note, total-DNA-PKcs [E6U3A] antibody (Cell Signaling Technology #38168) was validated by the manufacturer using knockdown in both western blotting and IF applications. To increase the confidence in the specificity of phospho-DNA-PKcs antibody in immunofluorescence we have conducted **new experiments** using DNA-PK knockdown (siRNA) in our ISPN model, The analysis using western blot and immunofluorescence is shown on **new Figure S6 B-C** of this revision. We performed staining of ISPNs after transfection (with either DNA-PKcs or scrambled control siRNA, **Fig. S6 C**). As shown, levels of pDNA-PKcs, detected with this antibody, were substantially lower after the knockdown, while levels of SC35 (used here as a reference protein) remained unaffected.

Comment

Fig 6D is also problematic since the PLA signal gives so my cytoplasmic signal, where you wouldn't expect these proteins to be. Standard controls for PLA experiments is the use of single antibody reactions so that these spurious signals can be understood and ruled out.

Response

We have performed technical negative single antibody controls within each experiment, and PLA site measurements were reported relative to these negative controls, as noted in the Methods section. We now have clarified it in corresponding Results sections. We have also included representative images showing each negative single antibody control for PLA assays (**new Figure S3A**).

Since we got very minimal appearance of PLA signals in either nucleus or cytoplasm using single antibody controls, we assume that cytoplasmic PLA sites detected with DNA-PK/HTT antibodies represent cytoplasmic proximity of these two proteins. Recently DNA-PKcs has been shown to localize not only in the nucleus but also in the cytoplasm and perform functions other than NHEJ. This is discussed in more detail in a new paragraph on p.9 of the revision. We have also observed some cytoplasmic localization of DN-PKcs in our ISPNs (Examples are shown on Fig. **S3 B**)

Comment

Figure 10e is misleading and should be removed

Response

Removed

Comment

Fig 11 adds little to nothing to the manuscript as presented. It is unclear where the data is coming from for the 4 conditions tested and it appears that this experiment may not have even worked for technical issues. It is unclear what additional value this experiment adds.

Response

We feel strongly that this data should be included: As described in the corresponding Results section (p12), APEX2 -based proximity labeling is a new approach that has only been previously reported once for HTT interactome, while most of the previous studies have used HTT pulldowns followed by mass spectrometry. Proximity-based proteomics provides several advantages -it can detect weak or dynamic interactions and can preserve the intracellular spatial information of the interactomes. This methods is not dependent on the antibodies to the bait protein eliminating potential bias. Furthermore, this approach confirmed enrichment of HTT interactome in the same categories (regulation of translation, RNA metabolism, chromatin remodeling) as were found by co-IP-MS method. In this revision we further emphasized that APEX2 study is preliminary and aimed towards method development for further analysis with endogenous tagging. (p. 4-last paragraph; p.12-last sentence)

Comment

Finally, a reduction in DNA damage signaling does not address the question of whether or not this is biologically significant. It was asked to look directly at NHEJ in these cells that display reduced DDR and this is still a valid experiment that should be performed prior to publication. If not, the significance of altered DDR and interaction with DNA-PKcs by HTT in 180CAG cells remains uncertain.

Response

Although we did not look at NHEJ efficiency directly (but plan to address this in the future studies), we established that double strand break (DSB) repair response is impaired in HD neurons which are more vulnerable to DSB induced stress (survival after bleomycin treatment). The text was revised (in the abstract, and discussion p.14) to acknowledge that the physiological outcome of HTT interaction with DNA-PKcs shown here remains to be examined.

February 24, 2026

RE: Life Science Alliance Manuscript #LSA-2025-03424RR

Dr. Tamara Ratovitski
Johns Hopkins University
600 North Wolfe Street
CMSC 5 South
Baltimore 21287

Dear Dr. Ratovitski,

Thank you for submitting your revised manuscript entitled "Huntingtin (HTT) interactome in regulation of DNA repair/remodeling and RNA processing pathways". We returned this to the new Reviewer 4, and as you will see below they have no further requests and recommend publication. We appreciate the care you have taken to address the concerns by this new reviewer, and we would be happy to publish your paper in Life Science Alliance pending final revisions necessary to meet our formatting guidelines.

MANUSCRIPT ORGANIZATION AND FORMATTING:

To avoid unnecessary delays in the acceptance and publication of your paper, please read the following information carefully. Full guidelines are available on our Instructions for Authors page, <https://www.life-science-alliance.org/authors>

- Please add an ORCID ID for the secondary corresponding author - they should have received instructions on how to do so.
- Please add the X and Bluesky handles of your host institute/organization, as well as your own and/or one of the authors, in our system.
- As your study uses human-derived iPS cells, please include an approval statement for the use of this material.
- In the methods section: Please provide details on PRM mass spectrometry rather than referring to a prior publication. Please indicate antibody concentrations used for both western blotting and immunofluorescence assays. Please describe bleomycin source and concentration used in the cell culture section. Please indicate the objective information (manufacturer, magnification, and NA), as well as z step sizes in the immunofluorescence section.
- The input blots for co-IP assays shown in Figure 4C and 8B appear to be duplicated (HTT and Tubulin). Please indicate this reuse in the legends for both of these figures.

LSA encourages authors to provide a 30-60 second video where the study is briefly explained. We will use these videos on social media to promote the published paper and the presenting author (for examples, see <https://docs.google.com/document/d/1-UWCfbE4pGcDdcgzcmiuJI2XMBJnxKYeqRvLLrLSo8s/edit?usp=sharing>). Corresponding or first-authors are welcome to submit the video. Please submit only one video per manuscript. The video can be emailed to contact@life-science-alliance.org

FINAL FILES:

The following items are required for acceptance.

The license to publish form must be signed before your manuscript can be sent to production. A link to the license to publish form will be available to the corresponding author only. Please take a moment to check your funder requirements.

Thank you for your attention to these final processing requirements. Please revise and format the manuscript and upload materials as soon as you are able.

Thank you for this interesting contribution to the literature. We look forward to publishing your paper in Life Science Alliance.

Sincerely,

Reviewer #4 (Comments to the Authors (Required)):

This revised manuscript now provides several new controls that help to increase the rigor of the study. While the authors did not perform key mechanistic studies that were requested, including experiments to validate the importance of HTT in NHEJ, this omission should not preclude publication in LSA. There still seems to be some typos in the manuscript, for example Metascape is misspelled in several places. In addition, in the future, text indicated the Figure number should be added to the Figures. The omission of these made it very difficult to find the data while reviewing this manuscript. Overall, this work presents an interesting new connection between HHT and the DDR, while also presenting several datasets that hopefully other researchers will find valuable in further understanding this disease and its connection with DNA damage and other cellular pathways.

March 4, 2026

RE: Life Science Alliance Manuscript #LSA-2025-03424RRR

Dr. Tamara Ratovitski
Johns Hopkins University
600 North Wolfe Street
CMSC 5 South
Baltimore 21287

Dear Dr. Ratovitski,

Thank you for submitting your Research Article entitled "Huntingtin (HTT) interactome in regulation of DNA repair/remodeling and RNA processing pathways". It is a pleasure to let you know that your manuscript is now accepted for publication in Life Science Alliance. Congratulations on this interesting work.

Your article will publish open access upon publication under a CC-BY license.

DISTRIBUTION OF MATERIALS:

Again, congratulations on a very nice paper. I hope you found the review process to be constructive and are pleased with how the manuscript was handled editorially. We look forward to future exciting submissions from your lab.

Sincerely,
